# Slow fluctuations in ongoing brain activity decrease in amplitude with ageing yet their impact on task-related evoked responses is dissociable from behavior

**Maria Ribeiro[1,2]\***, **Miguel Castelo-Branco[1,2]**

[1]CIBIT-ICNAS, University of Coimbra, Coimbra, Portugal; [2]Faculty of Medicine, University of Coimbra, Coimbra, Portugal

**Abstract** In humans, ageing is characterized by decreased brain signal variability and increased behavioral variability. To understand how reduced brain variability segregates with increased behavioral variability, we investigated the association between reaction time variability, evoked brain responses and ongoing brain signal dynamics, in young (N=36) and older adults (N=39). We studied the electroencephalogram (EEG) and pupil size fluctuations to characterize the cortical and arousal responses elicited by a cued go/no-go task. Evoked responses were strongly modulated by slow (<2 Hz) fluctuations of the ongoing signals, which presented reduced power in the older participants. Although variability of the evoked responses was lower in the older participants, once we adjusted for the effect of the ongoing signal fluctuations, evoked responses were equally variable in both groups. Moreover, the modulation of the evoked responses caused by the ongoing signal fluctuations had no impact on reaction time, thereby explaining why although ongoing brain signal variability is decreased in older individuals, behavioral variability is not. Finally, we showed that adjusting for the effect of the ongoing signal was critical to unmask the link between neural responses and behavior as well as the link between task-related evoked EEG and pupil responses.

**\*For correspondence:**
mjribeiro@fmed.uc.pt

**Competing interest:** The authors declare that no competing interests exist.

## Editor's evaluation

This paper examines an overlooked issue in research on the neurophysiology of cognitive aging – that the brain signals of older and younger subjects may differ due to physiological changes that bear little or no relevance to the cognitive processes being studied. Here, the authors demonstrate that older subjects exhibit reduced slow oscillations in EEG and pupil power and that controlling for these differences eliminated group differences in ERP and pupil dilation variability while strengthening their correlation with reaction time. The work should be of interest to those using EEG to study inter-individual and inter-group differences, particularly in the realm of aging and age-related disorders.

## Introduction

Brain signal variability holds valuable information about brain dynamics that should not be ignored when studying the link between neuronal activity and cognition. Brain signal variability can take many forms. Moment-to-moment variability of the ongoing signal reflects how much brain activity changes from one moment to the next, that is the brain activity range. Trial-by-trial variability in task-related evoked responses reflects differences in the brain responses to repeated task conditions, while brain signal entropy quantifies the irregularity of the time series.

Within-subject brain signal variability has been shown to change with ageing and is a robust marker of age (*Grady and Garrett, 2014*). The effect of ageing on brain signal variability has been studied using several measures, including temporal standard deviation (SD) and mean square successive difference (MSSD) of the blood-oxygen-level-dependent (BOLD) signal acquired with functional magnetic resonance imaging (fMRI) (*Grady and Garrett, 2014*; *Samanez-Larkin et al., 2010*), temporal SD of the spectral power within frequency bands measured with electroencephalography (EEG) (*Kumral et al., 2020*), entropy in EEG and magnetoencephalography (MEG) signals (*Kosciessa et al., 2020*; *McIntosh et al., 2014*; *Waschke et al., 2017*), the slope of the EEG power spectral density (PSD) (*Voytek et al., 2015*), and the consistency of task-related evoked responses (*Sander et al., 2012*; *Tran et al., 2020*). The temporal standard deviation of the BOLD signal and of the amplitude envelop of EEG frequency bands measured in the ongoing signal are markedly decreased in older adults (*Garrett et al., 2015*; *Grady and Garrett, 2018*; *Kumral et al., 2020*). Yet, the localization of these changes does not coincide suggesting that BOLD and EEG variability capture different aspects of brain function. In fact, age differences in BOLD SD might be explained by changes in cardiovascular and cerebrovascular factors that affect the link between neuronal activity and changes in blood flow (*Tsvetanov et al., 2021*). Moreover, BOLD MSSD, a measure of how much the BOLD signal changes across successive observations, is more often increased rather than decreased in older adults (*Boylan et al., 2021*; *Nomi et al., 2017*; *Samanez-Larkin et al., 2010*), highlighting the fact that age-related changes in signal dynamics can result in more or less 'signal variability' depending on the used metric. In fact, the relationship between the non-invasive measures of brain signal variability used in human neuroscience and neuronal noise is not trivial. Large signal fluctuations will increase signal standard deviation but might originate in a more stable and predictable signal associated with reduced noise. Higher signal variability is, therefore, not equal to a noisier signal. Consistent with this idea, EEG signal entropy increases with ageing and is associated with the EEG spectra slope (*Waschke et al., 2017*). Older adults present EEG signals with flatter spectra, i.e., more similar to white noise, potentially reflecting increased neuronal noise (*Voytek et al., 2015*). Higher brain noise is also consistent with the fact that older people present increased behavioral variability measured as less consistent reaction time across trials (*Grady and Garrett, 2018*; *MacDonald et al., 2012*). However, the link between brain signal variability, neuronal noise and behavioral variability remains to be clarified.

The ongoing brain signal (i.e. spontaneous brain activity) measured with fMRI or EEG during 'task-free', 'resting' periods reflects activity in large-scale networks each identified as a set of brain regions that show activity co-fluctuations, the resting-state networks (*Beckmann et al., 2005*; *Liu et al., 2017*). Notably, these networks are functionally active also during task performance (*Abreu et al., 2020*; *Smith et al., 2009*). The ongoing activation and de-activation of these networks presents a backdrop of activity on top of which task-related activity occurs (*Fox et al., 2006*). Not surprisingly, ongoing activity modulates task-related evoked responses that in turn affect task performance (*Becker et al., 2011*; *Fox et al., 2007*; *Mayhew et al., 2013*; *Ribeiro et al., 2016*). For example, fluctuations in pre-stimulus alpha power affect the amplitude of brain sensory responses, perception and reaction time (*Lou et al., 2014*; *Thut et al., 2006*; *van Dijk et al., 2008*). Older people display low-frequency (<12 Hz) brain signal fluctuations with reduced amplitude (*Kumral et al., 2020*), however, previous studies of the effect of ageing on the variability of task evoked responses have yielded contradictory results, revealing increased or decreased trial-by-trial consistency depending on stimulus or task characteristics (*Sander et al., 2012*; *Tran et al., 2016*; *Tran et al., 2020*; *Wiegand and Sander, 2019*). Finally, to our knowledge, no previous study has investigated the effect of ageing on the link between the dynamics of the ongoing brain activity, trial-by-trial variability of evoked responses and behavioral variability.

In this study, we aimed at clarifying the effect of ageing on brain signal dynamics and its impact on behavior by investigating the relationship between variability of ongoing brain signals, evoked responses, and behavior. We analysed EEG and pupil data from a previously published study acquired while a group of young and a group of older adults were engaged in a cued auditory go/no-go task (*Ribeiro and Castelo-Branco, 2019a*; *Ribeiro and Castelo-Branco, 2019b*; *Ribeiro and Castelo-Branco, 2021*). The EEG allows the non-invasive measurement of cortical electrical activity with high temporal resolution, while the pupillogram, acquired under conditions of constant luminance, is associated with fluctuations in arousal and reflects activity in the ascending brainstem neuromodulatory systems (*de Gee et al., 2017*; *Joshi et al., 2016*; *Murphy et al., 2014*; *Reimer et al., 2016*). Although,

activity modulation in subcortical arousal nuclei (reflected in changes in pupil size) correlates with the EEG signal (e.g. *Hong et al., 2014*; *Podvalny et al., 2021*), these two signals capture distinct aspects of brain function, both important and complementary for our understanding of the ageing brain. In our previous study, we showed that, in both signals, the cue stimulus evoked a preparatory response. In the EEG, the cue evoked a frontocentral negative potential [the contingent negative variation (CNV)] that increased in magnitude throughout the preparatory period. The CNV has been previously shown to correlate with reaction time on a trial-by-trial basis (*Boehm et al., 2014*), suggesting that trial-by-trial amplitude fluctuations in the CNV are associated with behavioral variability. In the pupillogram, the cue evoked a pupil dilation response of equal magnitude in both groups of participants (*Ribeiro and Castelo-Branco, 2019a*). In the current study, we investigated how trial-by-trial variability in the amplitude of these preparatory responses changed across age groups, how it related to behavior variability and how it was affected by ongoing brain signals. We found that, while behavioral variability was increased in older adults, this group of participants presented reduced variability of the evoked responses both in the EEG and in the pupillogram. We found that the decreased variability in the evoked responses emerged due to a reduction in the amplitude of the slow fluctuations observed in the ongoing signals and did not reflect reduced variability of the evoked responses per se. Once differences in the dynamics of the ongoing signals were accounted for, we observed no group differences in the variability of the evoked responses. Moreover, although the evoked responses were markedly affected by fluctuations in the ongoing signals, the amplitude modulations caused by the

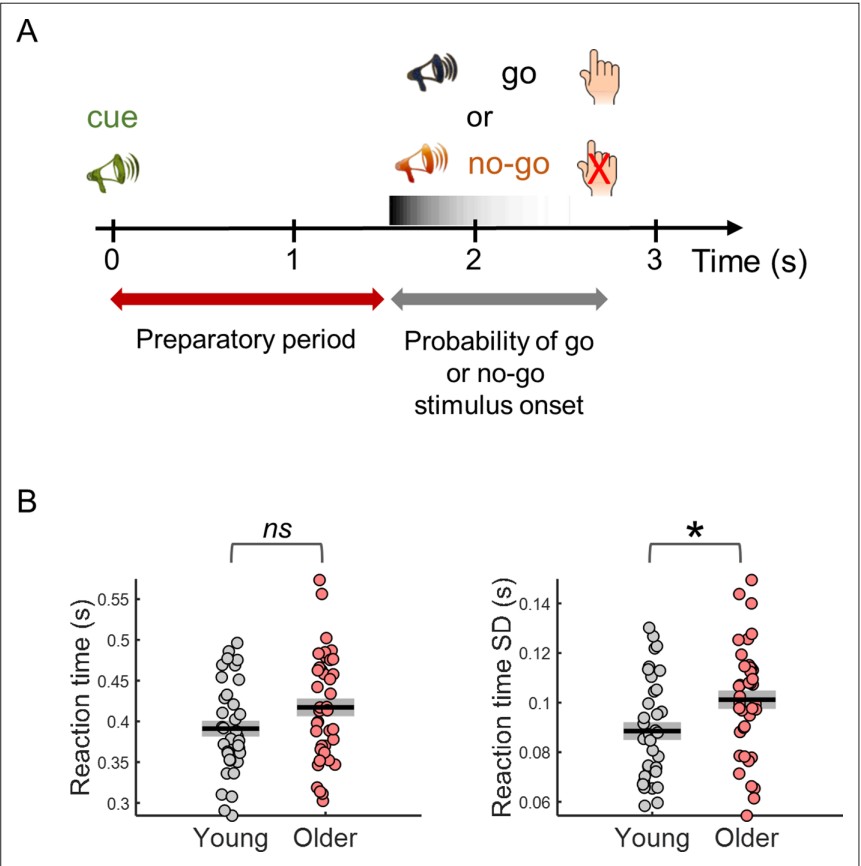

**Figure 1.** Behavioral task design and reaction time results. (**A**) Participants performed a cued auditory go/no-go task. The warning cue presented at the beginning of the trial was followed by a go or a no-go stimulus. Participants were instructed to respond with their right index finger as fast as possible upon detection of the go stimulus and to withhold the response in no-go trials. Auditory stimuli were pure tones of different frequencies (cue – 1500 Hz; go – 1700 Hz; no-go – 1300 Hz). (**B**) Median reaction time (left) and reaction time variability [across trials standard deviation (SD)] (right). Black horizontal line depicts mean across participants and grey box ± standard error of the mean. * p< 0.05; ns = not significant, in independent samples *t*-tests comparing across groups.

ongoing signal did not contribute towards behavioral variability, thereby explaining why although ongoing brain signal variability is decreased in older individuals, behavioral variability is not.

## Results

We analysed EEG and pupil data acquired while a group of young (N=36; age = 23 ± 3 years) and a group of older (N=39; age = 60 ± 5 years) adults were engaged in a cued-auditory go/no-go task (*Figure 1A*).

### Behavioral variability is increased in older adults

Reaction time variability was increased in older adults (*Figure 1B*). The older participants were on average slower responding to the go stimulus; however, this difference was not statistically significant [$t_{(73)}$=–1.84, p= 0.070]. Nevertheless, reaction time variability (within-subject across trials SD) was significantly higher in the older group [$t_{(73)}$=–2.53, p= 0.013]. As signal variability increases with signal mean, it is common to study the coefficient of variation (SD/mean) to estimate if the signal variability increased more than what would be expected from the increase in mean. Reaction time coefficient of variation was not significantly different across groups [$t_{(73)}$=–1.57, p= 0.120].

To evaluate task accuracy, we calculated the number of errors, including responses to the no-go stimulus (errors of commission), missed go trials, and responses to the cue (where participants responded to the cue instead of waiting for the go stimulus). Total number of errors were low (median of 2 and 4 in young and older groups, out of 120 trials). The number of errors of commission was not significantly different across groups (*U*=627, p=0.400; median of one error of commission across participants in each group, with 64% and 66% of young and older participants responding at least once to the no-go stimulus). In contrast, misses and responses to the cue occurred more often in the older group. Mann-Whitney tests revealed a significant effect of group for the number of misses (*U*=552, p=0.050, 22% of young participants presented at least one missed trial in comparison with 38% of older participants) and responses to the cue (*U*=538, p=0.022, 14% of young participants incorrectly responded to the cue stimuli at least once in comparison with 36% of older participants).

### Variability in the amplitude of preparatory evoked responses is decreased in the older group and is associated with reaction time

Trial-by-trial variability in the CNV amplitude (CNV variability) was decreased in the older group (*Figure 2*). During the preparatory period between the cue and the target, a frontocentral negative potential (the CNV) could be observed in the EEG of both groups of participants with strongest amplitude in the frontocentral electrode FCz (*Figure 2A and C*). Although the amplitude of this evoked response was on average stronger (more negative) in the older group, trial-by-trial variability (across trials standard deviation) was reduced in the older group (*Figure 2B and D*; see *Figure 2— figure supplement 1f* for single subject examples of trial-by-trial CNV variability). We observed significant group differences in the CNV amplitude and variability (*Figure 2E and F*). However, the group differences showed different topographic distributions. CNV amplitude presented strongest group difference in left central electrodes, while CNV variability presented strongest group difference in parieto-occipital electrodes.

Trial-by-trial CNV amplitude fluctuations were associated with reaction time fluctuations (*Figure 3*). Within-subject correlation analyses revealed a significant relationship between CNV amplitude and reaction time. Although, the correlation coefficients were on average smaller (closer to zero) in the young group, these did not differ significantly across groups (|*ts*|<2.6). The correlation values were significantly different from zero in a left frontocentral cluster of electrodes (one-sample *t*-tests including all participants; *Figure 3A*). As participants were responding with their right index finger the fact that the stronger correlation coefficients were observed in a left central cluster (contralateral to the response hand) suggests that the preparatory activity measured has a motor element. The CNV is comprised of different components including a motor-related late component (*Nagai et al., 2004*; *van Boxtel and Böcker, 2004*). This result suggests that the association between CNV amplitude and reaction time results from preparatory motor activity happening during that period.

Older individuals presented reduced trial-by-trial variability in cue-locked pupil dilation (PD) responses (*Figure 4*). The warning cue elicited a pupil dilation response that peaked around 3 s after

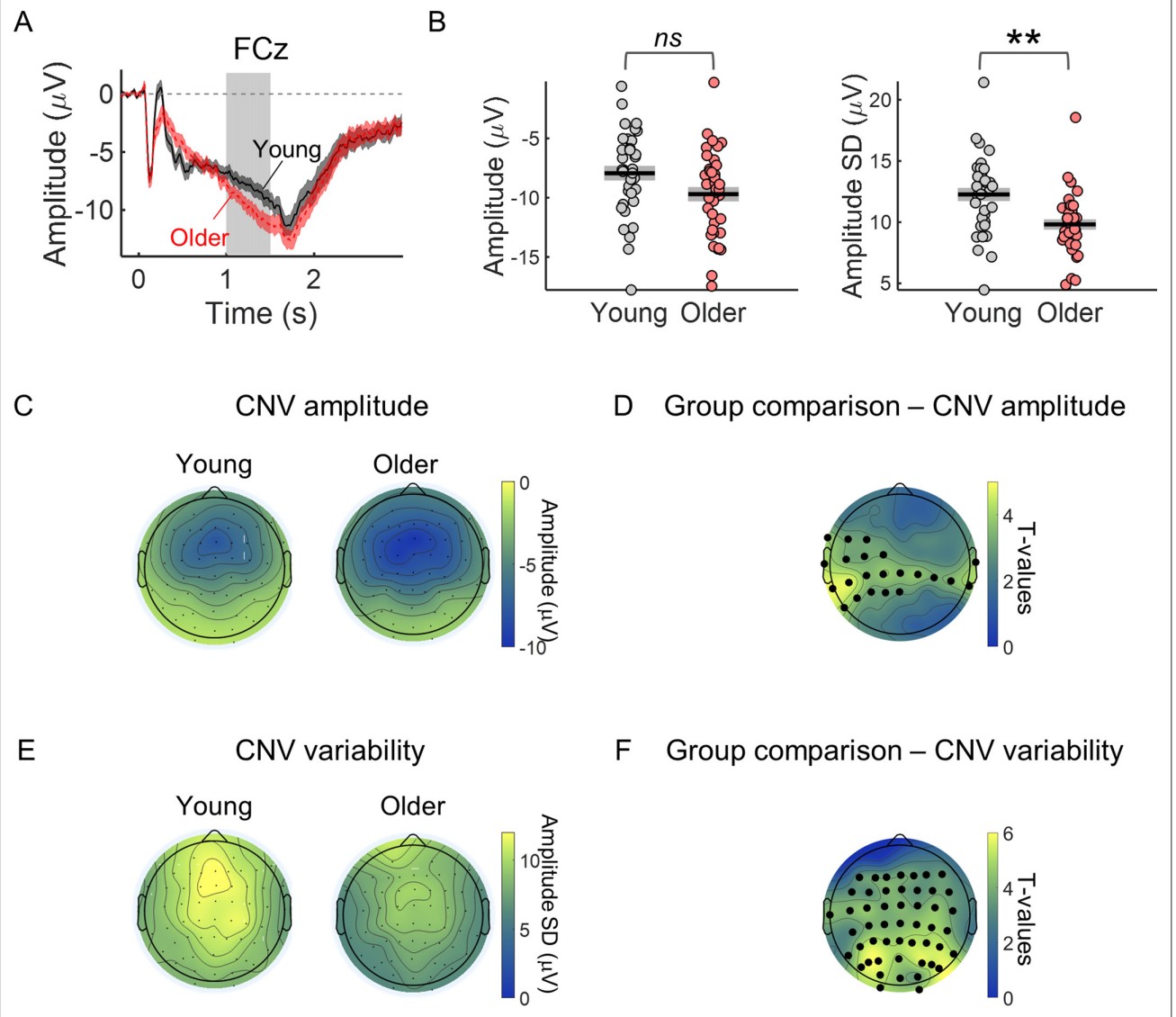

**Figure 2.** CNV variability is significantly decreased in the older group. (**A**) Cue-locked ERP measured in the electrode FCz (mean ± SEM across participants). Grey background highlights the time window used to average the ERP amplitude within each trial to study group differences in CNV amplitude and across trials CNV amplitude variability (1–1.5 s after cue onset). This time window was chosen to include the period of highest amplitude of the preparatory response while avoiding any activity related to target processing and therefore positioned just before the earliest target onset (1.5 s after cue onset). (**B**) Left, FCz CNV amplitude. Right, FCz CNV variability (across trials SD). Graphs depict individual data points (circles), mean (black horizontal line) and ± SEM across participants (grey box). ** p<0.01; ns = not significant, in independent samples *t*-tests comparing across groups after controlling for multiple comparisons across EEG channels. (**C**) Scalp topography depicting average CNV amplitude. (**E**) Scalp topographies of CNV variability. (**D and F**) Scalp topographies of *t*-values from independent samples *t*-tests comparing young and older participants at each electrode location. Correction for multiple comparisons was achieved using permutation tests and the 'tmax' method for adjusting the *p* values (*Blair and Karniski, 1993*; *Groppe et al., 2011*). Black circles highlight the electrodes where the group difference was significant after controlling for multiple comparisons (p<0.05).

The online version of this article includes the following figure supplement(s) for figure 2:

**Figure supplement 1.** Examples of cue-locked event-related potentials (ERPs) from young and older participants showing higher trial-by-trial variability in the young participants.

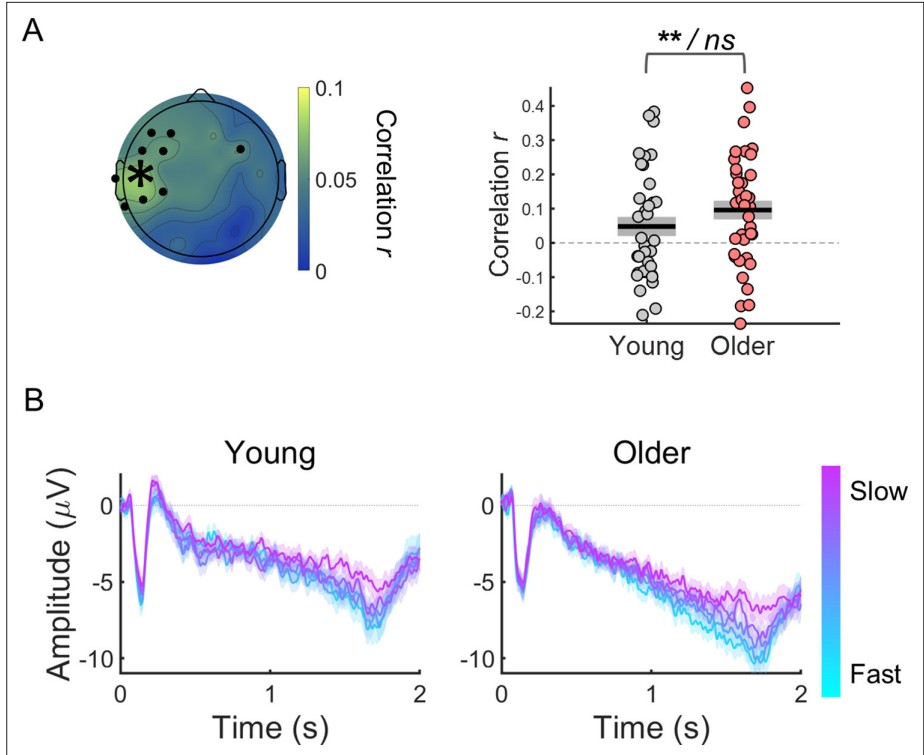

**Figure 3.** Within-subject correlation analyses between CNV amplitude and reaction time. (**A**) Scalp topographies of average correlation coefficients. Black circles highlight electrodes where correlation coefficients were significantly different from zero after controlling for multiple comparisons. Graph on the right shows individual data points (circles), mean (black horizontal line) and ± standard error of the mean across participants (grey box) of correlation coefficients for electrode C5 marked in the scalp topography with an asterisk. Statistical comparisons: one sample *t*-test including all participants (** p<0.01) / independent samples *t*-test comparing across groups (*ns*, not significant). (**B**) ERPs at electrode C5 divided within each participant in quintiles according to reaction time. Fast responses were associated with more negative CNV amplitudes in both groups. Data are represented as mean ± standard error of the mean across participants.

cue-onset (as reported before in *Ribeiro and Castelo-Branco, 2019a*; replotted in *Figure 4A*). PD response amplitude was not significantly different across groups [$t_{(71)}$=−1.82, p=0.073]. The variability of PD response amplitude was, however, significantly reduced in the older group [$t_{(71)}$=6.05, p<0.001; *Figure 4B*; see *Figure 4—figure supplement 1* for single subject examples of trial-by-trial variability in evoked responses].

Preparatory pupil dilation responses did not correlate with reaction time. Within-subject correlation coefficients were not significantly different across groups [$t_{(71)}$=1.10, p=0.276] or significantly different from zero [one-sample *t*-test including all participants, $t_{(72)}$ = 0.670, p=0.505].

In summary, trial-by-trial variability in the amplitude of the evoked responses was markedly reduced in older adults, while reaction time variability was increased. These observations suggest that the additional variability observed in the young group originates in a signal component that is not behaviorally relevant. In the following sections, we will delve into the origin of the variability in these evoked responses.

## Spectral properties of the ongoing EEG and pupil signals

Evoked neural responses occur on a background of ongoing neural activity that unavoidably adds to the responses measured (*Fox et al., 2006*; *Shimaoka et al., 2019*). It is possible therefore that age-dependent changes in the dynamics of ongoing activity explain the differences in the variability of task-related evoked responses. We explored the relationship between ongoing signal dynamics and the variability of the evoked responses by studying the spectral properties of ongoing EEG and pupil data and how these relate to the evoked responses.

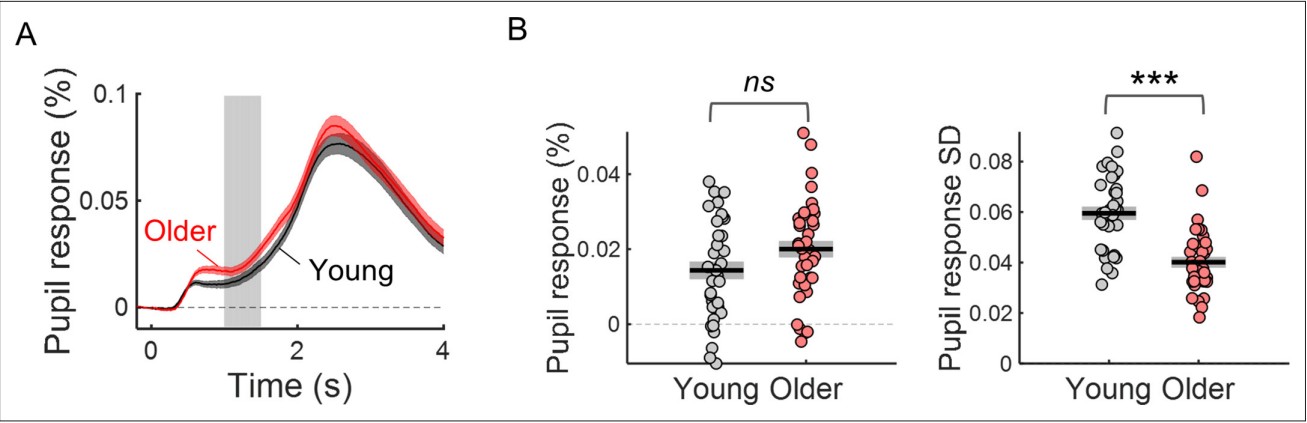

**Figure 4.** The variability of the cue-locked pupil dilation (PD) response is significantly reduced in older individuals. (**A**) Cue-locked pupil dilation response in young and older participants (mean ± SEM across participants). Grey background highlights time window (1–1.5 s after cue onset) where the PD amplitude was averaged within each trial to study the pupil dilation response elicited by the cue stimulus during the preparatory period before target onset. (**B**) Left, amplitude of PD response in young and older adults. Right, trial-by-trial variability in PD response (across trials standard deviation of PD amplitude) in young and older adults. Graphs depict individual data points (circles), mean (black horizontal line) and ± standard error of the mean across participants (grey box). *** p<0.001; ns = not significant, in independent samples *t*-tests comparing across groups.

The online version of this article includes the following figure supplement(s) for figure 4:

**Figure supplement 1.** Examples of cue-locked pupil dilation responses from young and older participants showing higher trial-by-trial variability in the young participants.

The ongoing EEG signal of older adults presented power spectral density (PSD) data that are less steep, with reduced offset (reduced amplitude at low frequencies), reduced alpha power and increased beta power (*Figure 5*). Ongoing EEG activity was estimated from the pre-stimulus period (3.5 s) just before cue onset, taking advantage of the long inter-trial intervals used. We analysed the PSDs using the FOOOF toolbox that separates the aperiod from the periodic component of the spectra (*Donoghue et al., 2020*). The aperiodic component can be described by its two parameters: the spectral exponent (a measure of how steep the spectrum is) and the offset (which reflects the uniform shift of power across frequencies). Both were significantly reduced in the older group (*Figure 5B*). Occipital alpha power was reduced in older individuals mainly in occipital channels. In contrast, beta power in frontal and central EEG channels was increased in the older group.

The PSDs of ongoing pupil signals of older individuals presented reduced exponent and offset (*Figure 6*). Ongoing pupil signal fluctuations happen on a slower time scale than the EEG fluctuations, therefore, it is important to analyse longer time periods to study their dynamics. To analyse ongoing pupil fluctuations in the absence of task related activity, we took advantage of the 4 min long pupillary recordings obtained at the beginning of the acquisition protocol while participants were fixating and passively listening to the cue stimulus being presented with the same frequency as in the task (*Ribeiro and Castelo-Branco, 2019a*). Besides fixating the centre of the screen no other overt task was required. Due to the absence of task or luminance changes, the dynamics of pupil fluctuations in these recordings can be mostly assigned to ongoing pupillary activity. We analysed the spectral content of these signals in 20 s epochs cut sequentially from these 4 min recordings (not locked to the auditory stimulus).The spectra of the ongoing pupillary signal followed a 1 /f distribution, with power decreasing with increasing frequency (*Figure 6*). The aperiodic parameters were significantly decreased in the older group [exponent: $t_{(70)}$ = 7.48, p<0.001; offset: $t_{(70)}$ = 4.17, p<0.001], that is the older group presented a flatter pupil spectrum with lower amplitude at the low frequencies. No obvious peaks were detected in the pupil spectra.

## Reduced variability in the evoked responses of older adults is linked to reduced amplitude in the slow fluctuations of ongoing activity

Spectral properties of ongoing signals are associated with trial-by-trial variability in evoked responses (*Figure 7*). We used correlational analyses to investigate if the dynamics of the ongoing EEG and pupil signals captured by the spectral parameters was associated with the variability in task-related

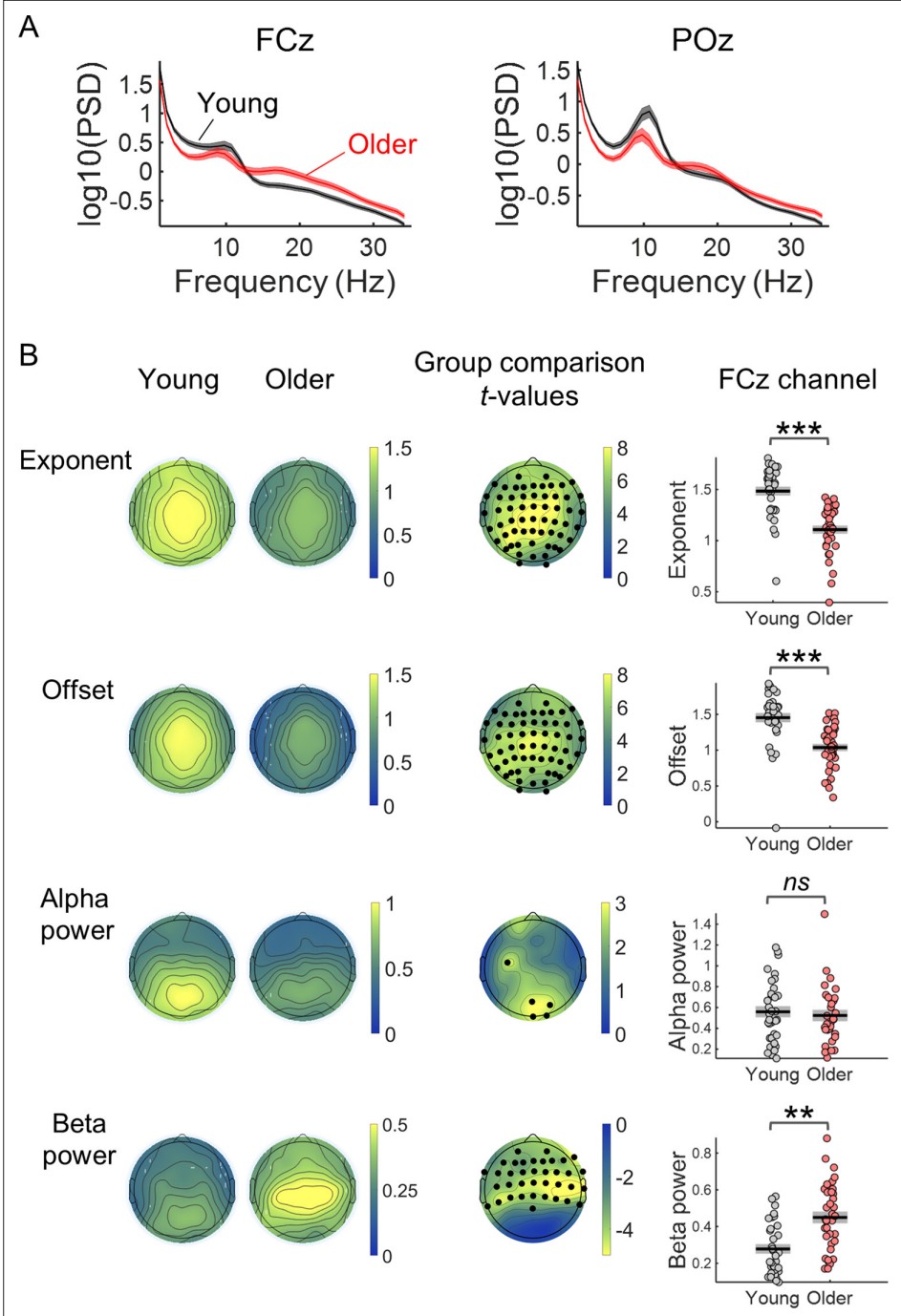

**Figure 5.** Comparison of the properties of the power spectral density (PSD) of the EEG pre-stimulus signal of young and older adults. (**A**) PSDs of the EEG signal in a window of 3.5 s before cue-onset measured in two EEG channels: the FCz, a frontocentral electrode where the preparatory CNV had maximal amplitude, and the POz, a parietocentral electrode where alpha oscillations were most prominent. Plots depict across participants mean ± standard error of the mean. (**B**) Scalp topographies of the average aperiodic parameters, exponent and offset, and alpha and beta power, extracted from the pre-stimulus PSD of each group of participants. Group differences for each of the parameters studied were estimated using independent t-tests. Channels that showed significant differences after controlling for multiple comparisons are highlighted in black. The graphs on the right show the exponent, offset, alpha power and beta power of the PSDs of young and older groups measured in the channel FCz. Graphs depict individual data points (circles), mean (black horizontal line) and ± standard error of the mean

*Figure 5 continued on next page*

*Figure 5 continued*

across participants (grey box). Participants where alpha or beta peaks were not detected were excluded from these graphs. **p<0.01; ***p<0.001; ns = not significant, in independent samples *t*-tests comparing across groups.

The online version of this article includes the following figure supplement(s) for figure 5:

**Figure supplement 1.** Goodness-of-fit of FOOOF model fitting to EEG spectral data was significantly different across groups and correlated with the model aperiodic parameters, exponent and offset.

evoked responses across participants. For the EEG, to facilitate data presentation and interpretation of the following analyses, we focused on the FCz channel where the CNV showed its highest amplitude. Similar findings were observed in the other EEG channels, including the ones presenting significant correlations with behavior (*Figure 7B* and *Figure 7—figure supplement 1*). Participants with higher spectral offset and exponent presented higher CNV and PD variability (*Figure 7*). Alpha power presented a positive association with CNV variability only in the young group and in posterior EEG channels (*Figure 7B*). Beta power was not associated with CNV variability. These findings suggest that the aperiodic signal fluctuations are intrinsically linked to the variability observed in the evoked responses. We then sought to determine which frequencies in the aperiodic spectra were more predictive of the variability in the amplitude of the evoked responses. We found that the correlation between variability in the amplitude of the evoked responses and the amplitude of the power spectra was particularly strong in the lower frequencies (*Figure 7—figure supplement 1*). Importantly, the correlations were significant in both the aperiodic component of the fitted (FOOOF) spectra and in the raw (total) power spectra. In the total spectral power, the correlation between PD variability and spectral power was highest around 0.44 Hz and decreased for lower frequencies. In the EEG analyses, correlation was highest at 1 Hz, the lowest frequency analysed. These findings suggest that slow aperiodic fluctuations observed in the ongoing signals are associated with trial-by-trial variability in the amplitude of the evoked responses, both in the EEG and in the pupillogram.

The group differences in CNV and PD variability were markedly reduced once the effect of the spectral exponent, offset or slow spectral power were regressed out (*Figure 8*). This was evident in the significant interaction effects observed in repeated measures ANOVAs including as within-subject factor adjustment for confounds (with and without adjustment) and group as between-subject factor [CNV variability adjustment x group interaction effect when adjusting for exponent $F_{(1, 72)}$=47.2, p<0.001, offset $F_{(1, 72)}$=14.3, p<0.001, or slow spectral power $F_{(1, 70)}$=13.4, p<0.001; PD variability adjustment x group interaction effect when adjusting for exponent $F_{(1, 71)}$=54, p<0.001, offset $F_{(1, 71)}$=19, p<0.001, or slow spectral power $F_{(1, 71)}$=62, p<0.001]. In fact, after adjusting for the confounds,

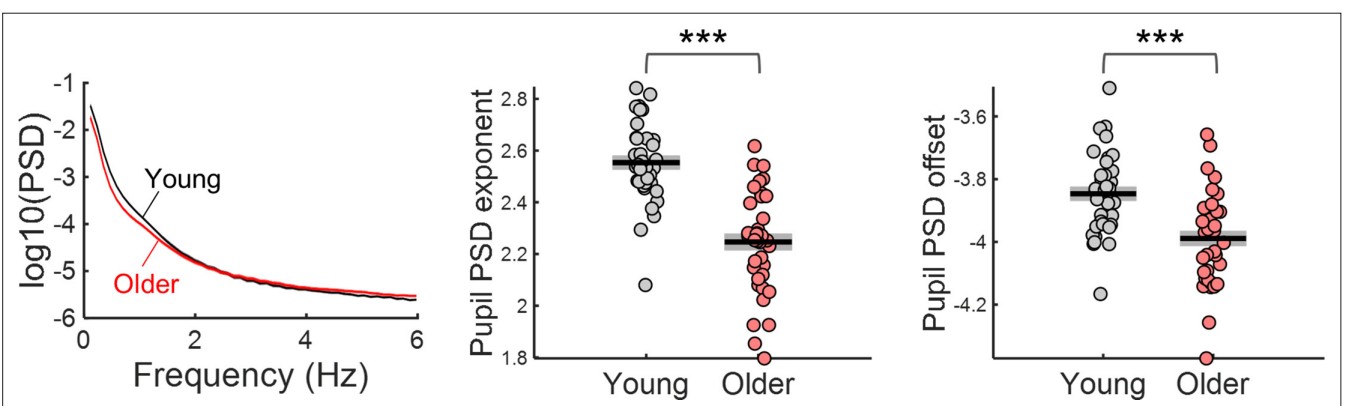

**Figure 6.** Power spectral density (PSD) of ongoing pupil signals acquired during passive fixation under constant luminance conditions. Left, PSD graph represents across participants mean ± standard error of the mean of the young and older group. Centre and right, aperiodic parameters exponent and offset of young and older participants. Graphs depict individual data points (circles), mean (black horizontal line) and ± standard error of the mean across participants (grey box). *** p-value < 0.001 in independent samples *t*-tests.

The online version of this article includes the following figure supplement(s) for figure 6:

**Figure supplement 1.** Goodness-of-fit of FOOOF model fitting to spectral pupil data, *R* (transformed into Fisher's Z) was similar across groups [independent *t*-test: $t_{(71)}$ = 1.45, p=0.151].

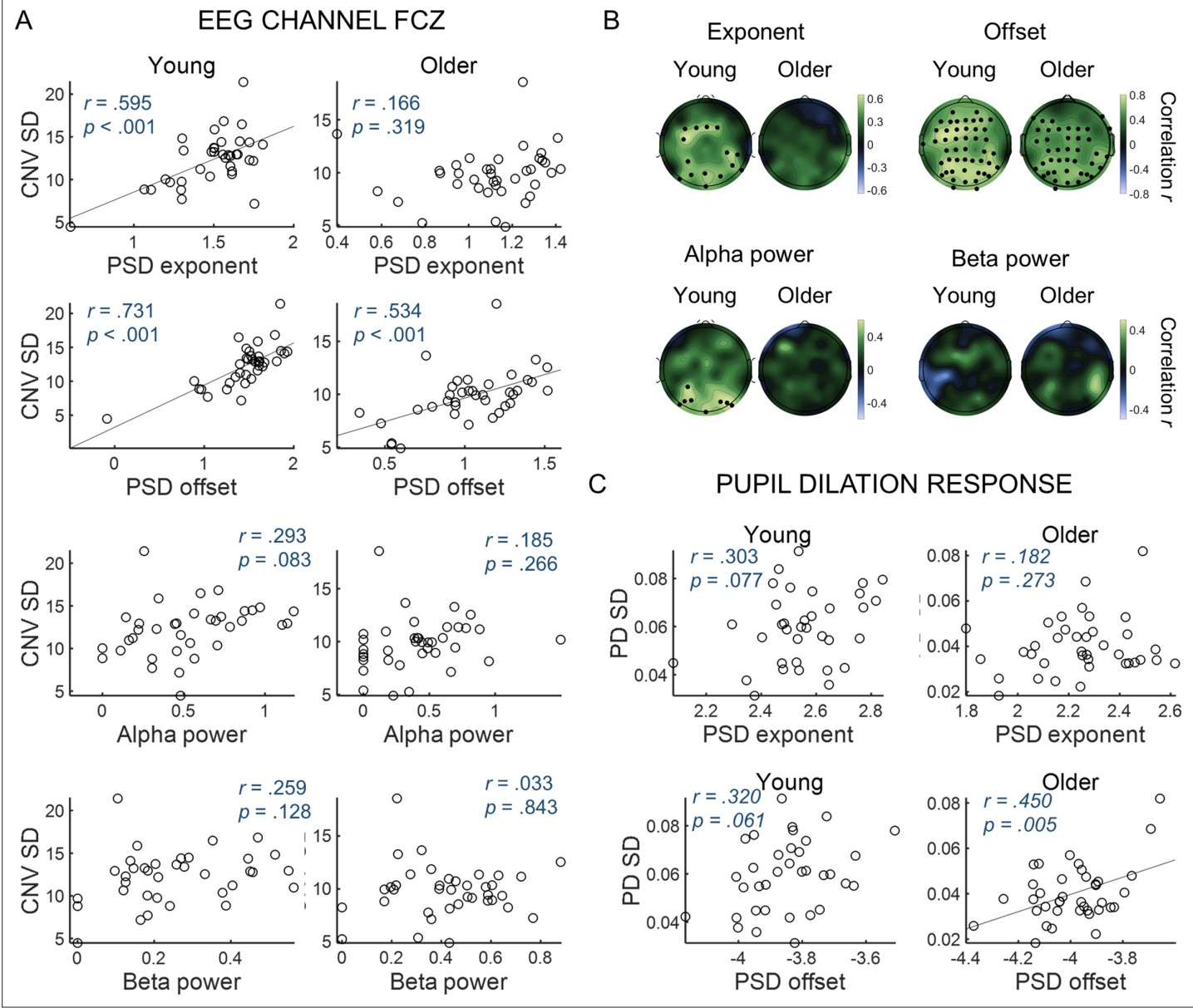

**Figure 7.** Variability in the amplitude of the evoked responses correlates with parameters of the power spectral density (PSD) of the ongoing signals. (**A**) CNV variability measured in the FCz channel plotted against (from top to bottom) exponent, offset, alpha power or beta power estimated from the PSDs of ongoing pre-stimulus EEG signal measured in the same EEG channel. The correlation *p*-values shown were not corrected for multiple comparisons. (**B**) Scalp topographies of correlation coefficients between CNV variability and exponent, offset, alpha power or beta power estimated from the PSDs of ongoing pre-stimulus EEG signal. Black circles highlight electrodes where the correlations were significant after controlling for multiple comparisons. (**C**) Variability in the pupil dilation (PD) response [PD standard deviation (SD)] plotted against the aperiod parameters of the PSD of the ongoing pupil signal, exponent and offset.

The online version of this article includes the following figure supplement(s) for figure 7:

**Figure supplement 1.** Power spectral densities (PSD) of ongoing signals correlate with variability in the evoked responses.

the CNV variability was no longer significantly different across groups [adjusting for exponent $t_{(72)}$ = 0.661, p=0.511; adjusting for offset $t_{(72)}$ = 0.243, p=0.809; adjusting for slow spectral power $t_{(72)}$ = 0.979, p=0.331; *Figure 8*]. Similar findings were observed in the other EEG channels after controlling for multiple comparisons. After adjusting for the exponent, group differences in CNV variability were observed only in two posterior EEG channels (PO8 and CB2). No channels showed group differences after adjusting for offset or slow spectral power. The group difference in the PD variability was also

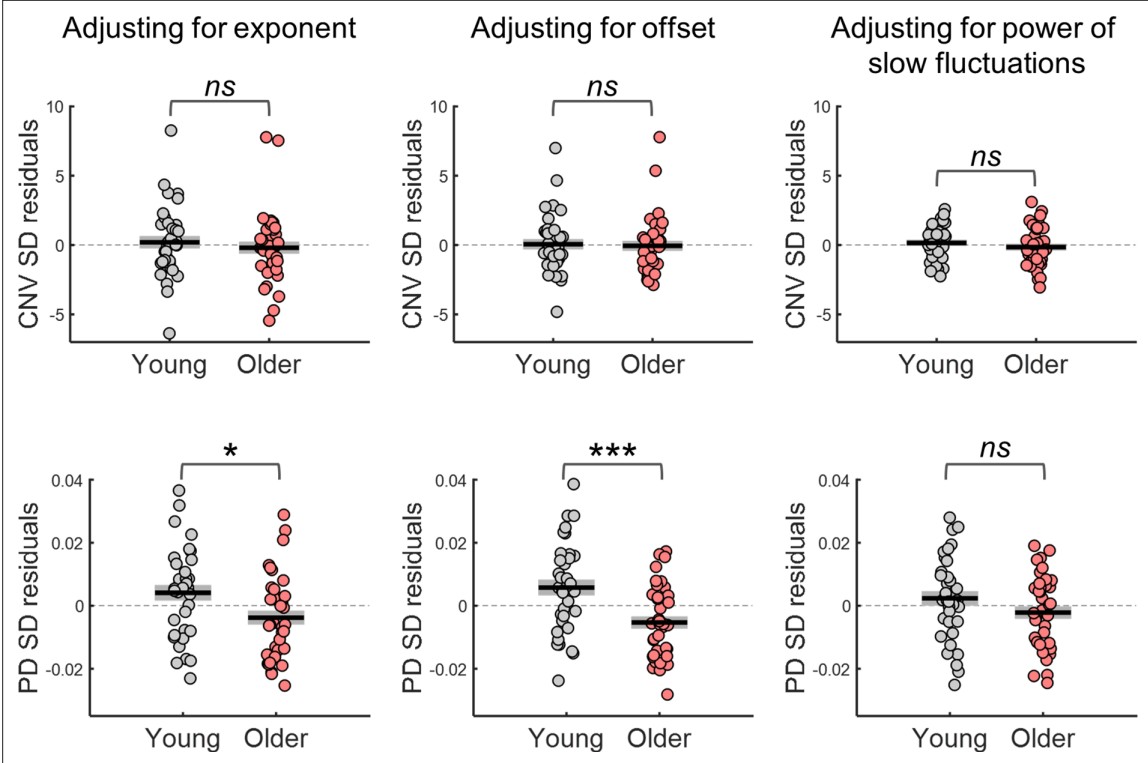

**Figure 8.** Variability in the amplitude of the evoked responses after adjusting for differences in the parameters of the power spectral density (PSD) of the respective ongoing signals. Top, variability in CNV amplitude measured at electrode FCz; bottom, variability in pupil dilation (PD) responses. Left, adjusted for the PSD's exponent; centre, adjusted for the PSD's offset; right, adjusted for the power of the slow fluctuations (power at 1 Hz in the EEG and power at.44 Hz in the pupil signal). Graphs depict individual data points (circles), mean (black horizontal line) and ± standard error of the mean across participants (grey box). * p<0.05; *** p<0.001; ns = not significant, in independent samples *t*-tests comparing across groups.

completely explained by the power of slow fluctuations observed in the ongoing pupillary signal [$t$-test group comparison no longer significant: $t_{(71)}$ = 1.54, p=0.129; *Figure 8*]. However, adjusting for spectral exponent or offset did not completely explain the group difference [adjusting for exponent $t_{(71)}$ = 2.45, p=0.017; adjusting for offset $t_{(71)}$ = 3.65, p<0.001]. These findings indicate that the age-related reduction in the power of the slow aperiodic fluctuations underlie the group differences in trial-by-trial variability in the amplitude of the evoked responses.

## Factors that contribute to trial-by-trial variability in evoked responses

There are two possible mechanisms through which background signal fluctuations affect the amplitude of the evoked responses. Background signal fluctuations might reflect fluctuations in brain state and different brain states will be associated with different evoked responses; and/or background signal fluctuations and the evoked responses are two independent signals that sum giving rise to a composite signal where the evoked response appears on top of a fluctuating baseline thereby modulating its shape and amplitude - a simple additive mechanism. In this latter case, the phase of the background fluctuations at the cue-onset rather than the instantaneous amplitude would better capture the relationship between the measured evoked responses and the ongoing signal (at the same amplitude values, the phase of the oscillation will define if the signal is increasing or decreasing). We explored this relationship by estimating the phase of the slow fluctuations in the EEG and pupil signals at cue-onset. For illustrative purposes, we plotted the relationship between the pre-stimulus phase and the amplitude of the respective evoked responses (*Figure 9*). The evoked responses showed a dependence on the phase of the ongoing fluctuations in line with what would be expected if the measured evoked responses were the summation of an evoked response with an ongoing fluctuating signal (*Figure 9B and C*). This relationship was also different from what would be expected if the evoked responses were affected solely by the amplitude of the baseline signal. This is further explored quantitatively in the next paragraph.

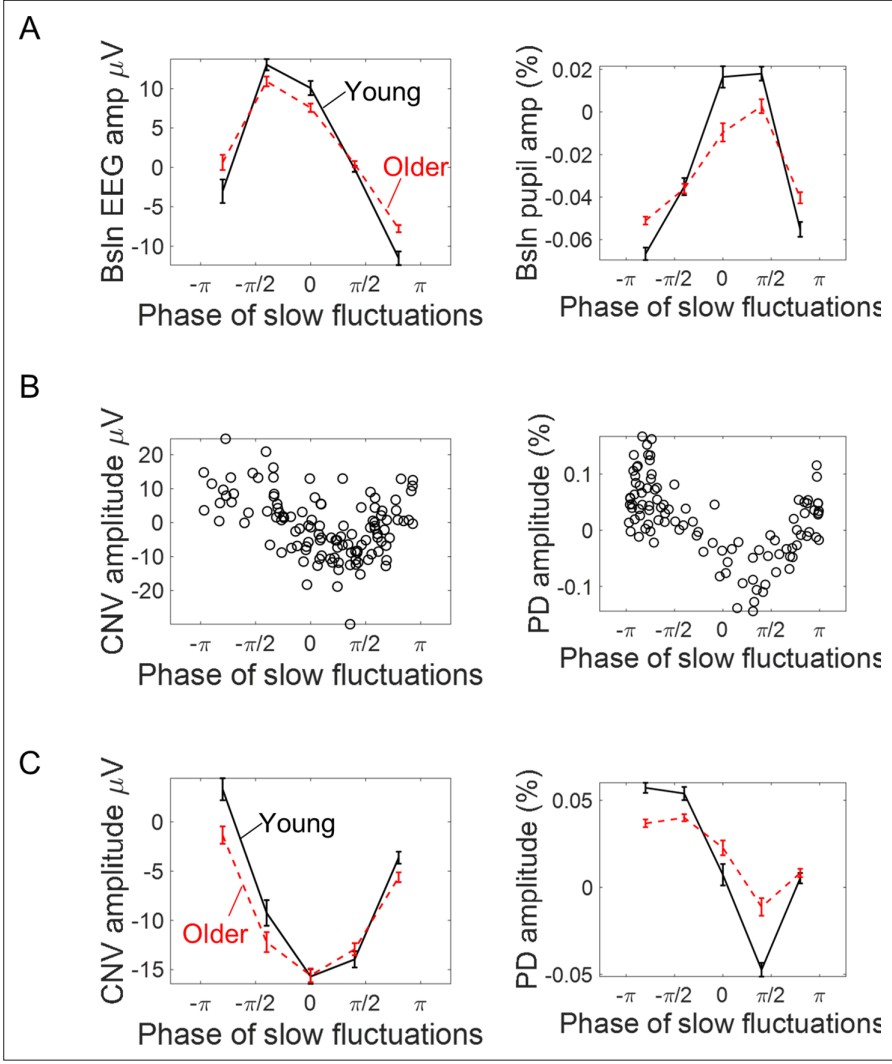

**Figure 9.** Association between the phase of slow ongoing fluctuations at cue-onset and the amplitude of the evoked responses in the EEG (left) and pupil (right) signals. (**A**) Baseline (pre-cue) signal amplitude sorted accordingly to the estimated signal phase (mean ± standard error of the mean across participants). As expected, signal amplitude reaches its peak at zero phase and its minimum at ± π. (**B**) Example data from two participants showing how the single trial amplitude of the evoked responses changed with the phase of the ongoing slow fluctuations. (**C**) Amplitude of the evoked responses split into quintiles (within each participant) according to the phase of the ongoing signals at cue-onset (mean ± standard error of the mean across participants).

Ongoing signal fluctuations strongly affected the amplitude of the evoked responses (*Figure 10*). We quantified the association between EEG/pupil phase of slow fluctuations estimated at cue-onset (θ) and the amplitude of the respective evoked responses using linear regression. This relationship can be captured in a linear regression by representing the phase as a pair of variables: the cosine and the sine (*Nurhab et al., 2017*). We multiplied the cosine and sine functions by the amplitude envelop (r) of the slow signal fluctuations because the amplitude envelop varies throughout the recordings and these fluctuations will have differential effects on the evoked responses depending on their amplitude. The two predictors associated with the slow ongoing signal fluctuations were $r \bullet sin\ \theta$ and $r \bullet cos\ \theta$. Note that $r \bullet cos\ \theta$ is equal to the signal amplitude and therefore to the amplitude at baseline, while $r \bullet sin\ \theta$ will capture the additional effect of the signal phase. To study how these factors affected the trial-by-trial amplitude of the task-related responses, we run generalized linear regression models (GLMs) for each participant and separately for each signal type - EEG and pupil. We included as dependent variable the amplitude of the evoked response and as predictors $r \bullet sin\ \theta$, $r \bullet cos\ \theta$ and, additionally, time-on-task (that can lead to differential responses due to learning, adaptation,

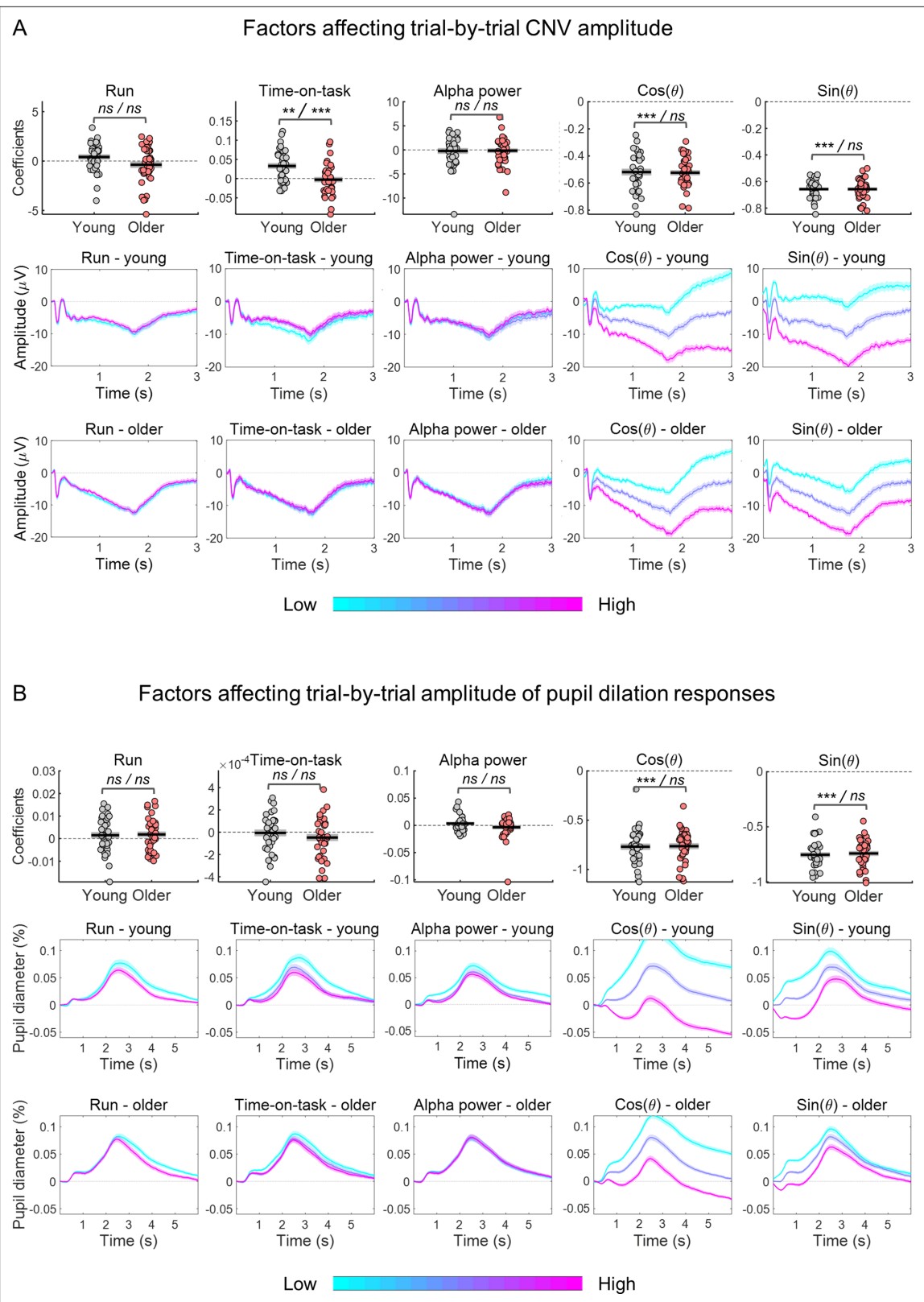

**Figure 10.** Estimated coefficients from within-subject generalized linear regression models (GLMs) including as response variable the amplitude of the evoked response and as predictors the effects of time-on-task, pre-stimulus alpha power and the cosine and sine of the phase at cue-onset of the ongoing slow fluctuations. Run was included as categorical predictor. (**A**) Effect of predictors on CNV amplitude measured at FCz. (**B**) Effect of predictors on pupil dilation response (PD) amplitude. (**A and B**) Top, graphs depicting estimated coefficients (circles = individual data points; black horizontal line

*Figure 10 continued*

= across participants mean; grey box = ± standard error of the mean across participants). Middle and bottom, cue-locked evoked responses divided in tertiles (within each participant) sorted according to each of the predictors (mean ± standard error of the mean across participants). Note, that the graphs show raw responses not adjusted for the other predictors. Statistical comparisons: one sample *t*-test including all participants / independent samples *t*-test comparing across groups - ** p<0.01; *** p<0.001; ns = not significant.

or increased fatigue), and pre-stimulus alpha power (that reflects moment-to-moment spontaneous fluctuations in brain state related to alertness or attentional fluctuations; *van Diepen et al., 2015*). Run was included as categorical predictor (two consecutive runs of 8 min each were acquired). These models explained a large amount of the evoked responses variance ($R^2$ mean ± SE: EEG young 67% ± 8%; EEG older 63% ± 8%; pupil young 73% ± 1%; pupil older 67% ± 1%). The explained variances were significantly decreased in the alternative model without the $r \bullet sin\ \theta$ predictor that captured the effect of the ongoing signal phase [$R^2$ mean ± SE without $r \bullet sin\ \theta$: EEG young 27% ± 11%; EEG older 28% ± 10%; pupil young 40% ± 1%; pupil older 37% ± 1%, paired *t*-test comparing the two models' explained variance: EEG $t_{(73)}$ = –26.8, p<0.001; pupil $t_{(71)}$ = –22.9, p<0.001], suggesting that the effect of the ongoing signal on the amplitude of the evoked responses is best captured by including amplitude and phase of the ongoing signal at baseline. We tested the estimated coefficients against zero using one-sample *t*-tests. The EEG GLM revealed that the CNV amplitude depended on time-on-task [it became less negative with time-on-task; $t_{(73)}$ = 2.84, p=0.006; *Figure 9A*], and on the phase of the ongoing fluctuations at cue-onset captured by the cosine [$t_{(73)}$=–36.3, p<0.001] and the sine [$t_{(73)}$=–81.7, p<0.001] functions (*Figure 10A*). It was independent of run [$t_{(73)}$=0.035, p=0.973] and pre-stimulus alpha power [$t_{(73)}$=–0.530, p=0.598]. Time-on-task had a stronger effect on the evoked responses of the young group in comparison with the older group [independent samples *t*-test: $t_{(72)}$ = 3.74, p<0.001]. The other coefficients did not present significant effects of group [run: $t_{(72)}$ = 1.98, p=0.051; pre-stimulus alpha: $t_{(72)}$ = –0.098, p=0.923; cosine: $t_{(72)}$ = 0.189, p=0.851; sine: $t_{(72)}$ = –0.007, p=0.994]. The pupil GLM revealed that the only predictors that significantly contributed towards trial-by-trial PD variability were the cosine and the sine of the pre-stimulus phase [one-sample *t*-test testing estimated coefficients against zero - cosine: $t_{(71)}$ = –38.2, p<0.001; sine: $t_{(71)}$ = –51.3, p<0.001]. The other predictors were not significantly different from zero [run: $t_{(71)}$ = 1.88, p=0.064; time-on-task: $t_{(71)}$ = –1.34, p=0.185; alpha power: $t_{(71)}$ = –0.061, p=0.952]. None of the coefficients showed a significant group difference [run: $t_{(70)}$ = –0.198, p=0.844; time-on-task: $t_{(71)}$ = –1.00, p=0.321; alpha power: $t_{(70)}$ =

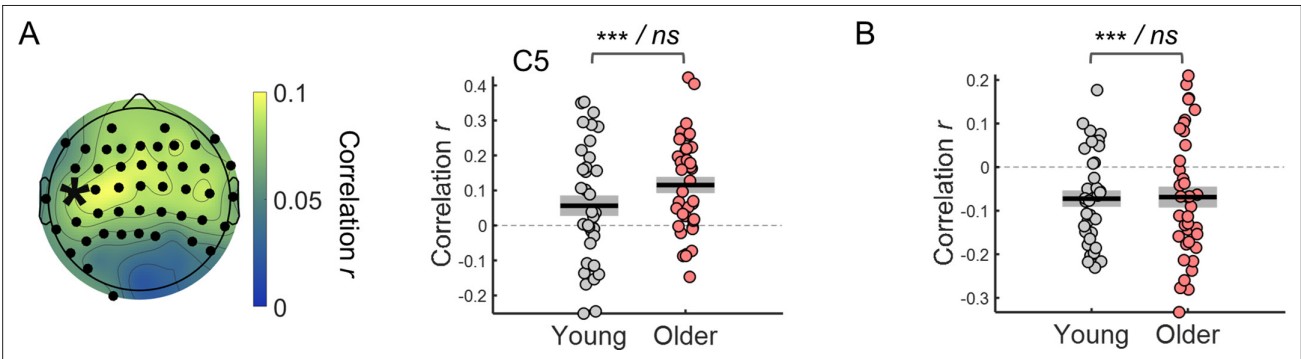

**Figure 11.** Within-subject correlation between single trial amplitude of the preparatory evoked responses adjusted for the effect of the ongoing signal fluctuations and reaction time. (**A**) Left, scalp topographies of correlation coefficients. Electrodes where the correlation coefficients were significantly different from zero after controlling for multiple comparisons are highlighted in black. Right, example of correlation coefficients between the CNV amplitude adjusted for ongoing signal phase fluctuations measured in the channel C5 (marked with an asterisk in the scalp topography) and reaction time. (**B**) Correlation coefficients between the amplitude of the pupil dilation responses adjusted for the phase of the ongoing signal fluctuations and reaction time (circles = individual data points; black horizontal line = across participants mean; grey box = ± standard error of the mean across participants). Statistical comparisons: one sample *t*-test including all participants / independent samples *t*-test comparing across groups - *** p<0.001; ns = not significant.

The online version of this article includes the following figure supplement(s) for figure 11:

**Figure supplement 1.** After regressing out the effect of ongoing slow fluctuations the variability of the evoked responses was still significantly reduced in the older group.

1.68, p=0.098; cosine: $t_{(70)}$ = –0.143, p=0.887; sine: $t_{(70)}$ = –0.396, p=0.693]. These results suggest a strong effect of the amplitude and phase of the ongoing signal fluctuations on the evoked responses (*Figure 10A and B*), consistent with the idea that there is a drift in the signal baseline that adds to the evoked responses pushing them towards positive or negative values depending on the phase at cue-onset.

To test if the variability in the evoked responses associated with the ongoing signal fluctuations was behaviorally relevant, we studied how adjusting for the signal phase at baseline changed the correlation between the evoked responses and reaction time. After regressing out the effect of the ongoing signal fluctuations from the amplitude of the evoked responses, these were associated with reaction time at least as strongly as without the adjustment. In particular, the CNV amplitude residuals correlated with reaction time in a larger number of electrodes than without the adjustment and unmasked a significant relationship between the CNV amplitude and reaction time in frontal electrodes that was not observed without the adjustment (*Figure 11A*), suggesting that the relationship between CNV amplitude and behavior is independent of the baseline fluctuations. On average the correlation coefficients with the adjustment were slightly more positive than without the adjustment. This difference was significantly different in three EEG channels (F4, C1, and C4) after correcting for multiple comparisons. For the pupil responses, we found that regressing out the effect of ongoing signal fluctuations was crucial to detect the link between PD evoked responses and reaction time, that is the PD amplitude residuals correlated with reaction time [*Figure 11C*; one-sample *t*-test: $t_{(72)}$ = –4.77, p<0.001]. Notably, the correlation coefficients with the adjustment for the phase of the ongoing signal were significantly different from the coefficients without the adjustment [paired *t*-test comparing with and without adjustment: $t_{(72)}$ = –5.08, p<0.001]. These findings suggest that the effect of the ongoing signal on the evoked responses is not behaviorally relevant and can mask brain-behavior associations.

It is important to note that after regressing out the effect of the ongoing signal's phase at cue-onset, the variability of the evoked responses was still significantly reduced in the older group (*Figure 11—figure supplement 1*), suggesting that the baseline (pre-stimulus) values are not enough to completely predict the effect of the ongoing signal fluctuations on the evoked responses in line with the idea that these slow signal fluctuations are aperiodic and therefore not completely predictable (*Shimaoka et al., 2019*).

## Joint contribution of pupil dilation and CNV responses to reaction time variability

After adjusting the evoked responses for the effect of the ongoing slow signal fluctuations, both the pupil dilation and the CNV responses correlated with reaction time. However, it was not clear if and how their contributions to the behavioral variability interacted. To address this question, we first checked if these two responses correlated with each other. The pupil dilation response shows a delay of hundreds of milliseconds in relation to the neural responses with which it is associated (*Joshi et al., 2016*; *Joshi and Gold, 2022*; *Reimer et al., 2016*; *Yang et al., 2021*). To take into account the delay in the pupil response, we correlated the CNV amplitude measured in the time window at the end of the preparatory period (1–1.5 s after cue onset) with the pupil dilation response measured in three time windows: one concomitant with the CNV amplitude measurements as used in the previous analyses (1–1.5 s after cue onset), and two later time windows (1.5–2 s and 2–2.5 s after cue onset). In within-subject correlation analyses, we found that the single trial measurements of the CNV and pupil responses were significantly correlated with stronger correlation coefficients observed in posterior channels (*Figure 12A and B*). The correlation coefficients were stronger when the pupil response was measured 500ms after the CNV measurements. Just like for the correlations with task performance, these correlations were stronger after adjusting for the ongoing slow signal fluctuations. The effect of pupil time window was stronger in the older group and the correlation coefficients showed a stronger effect of adjustment for the later time windows. Statistically, in a repeated measures ANOVA including as within-subject factors the effect of adjustment and pupil time window, and group as between-subjects factor, across channels average correlation coefficients presented a significant effect of pupil time window [$F_{(2, 70)}$=9.43, p<0.001 ], a significant effect of adjustment [$F_{(1, 70)}$=4.59, p=0.036 ], a significant interaction pupil time window x group [$F_{(2, 70)}$=3.26, p=0.041], and a significant interaction pupil time window x adjustment [$F_{(2, 140)}$=7.890, p<0.001]. The across channels average correlation coefficients showed no effect of group [$F_{(1, 70)}$=1.17, p=0.283], and no group x adjustment interaction

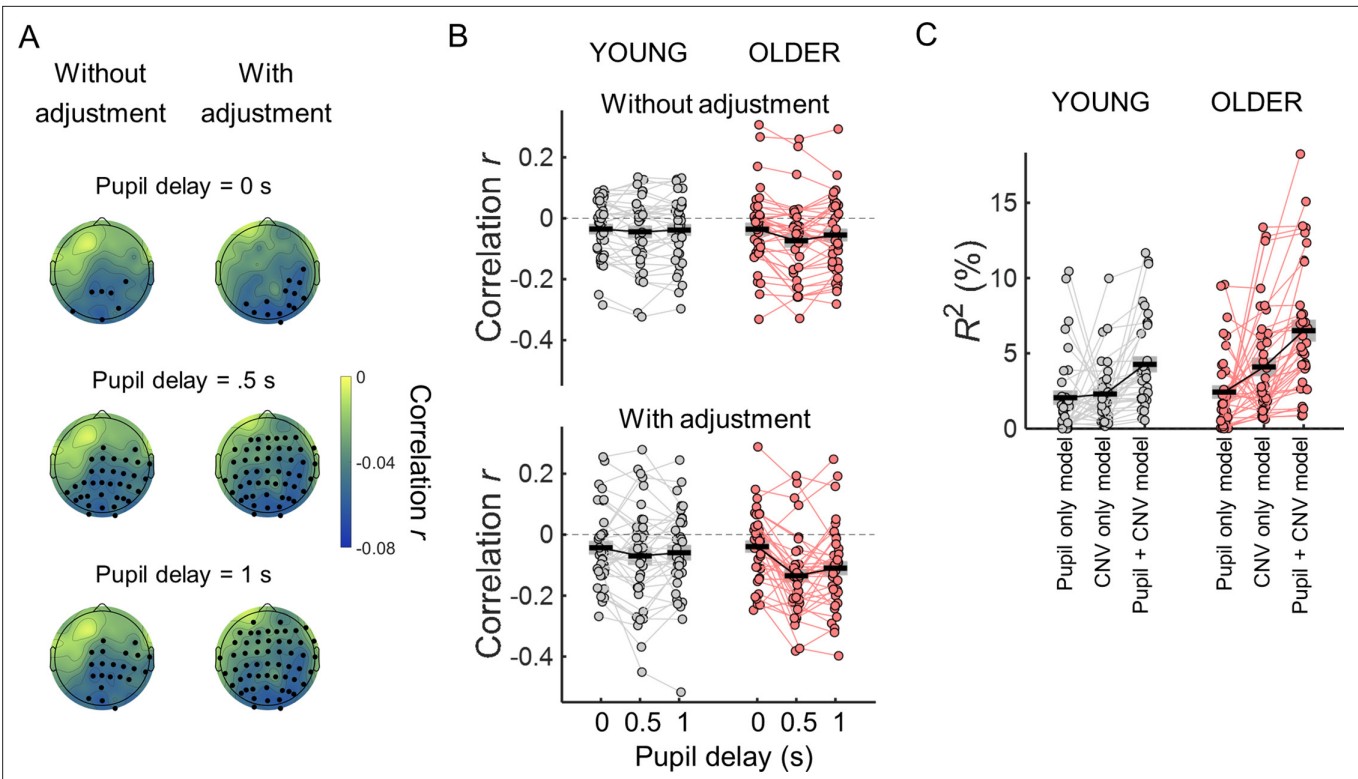

**Figure 12.** The CNV and the pupil response are correlated, nevertheless each contributes independently towards reaction time. (**A**) Scalp topographies of within-subject correlation values between single trial CNV amplitude values and single trial pupil dilation (PD) amplitude values (across participants average correlation coefficients including all participants) for the three time windows used to calculate pupil dilation average amplitude. Correlation coefficients were estimated using the 'raw' amplitude of the preparatory evoked responses (left), or after adjusting the amplitude of the evoked responses for the effect of the ongoing slow signal fluctuations (right). EEG channels where the correlation coefficients were significantly different from zero after controlling for multiple comparisons are highlighted in black. (**B**) Across channels average correlation coefficients between single-trial CNV and pupil responses for both age groups for the three pupil time windows and adjusting or not the signals for the effect of the ongoing slow signal fluctuations. (**C**) Total variance explained (**R**$^2$) by the multiple regression models including reaction time as dependent variable and as independent variables CNV and/or PD residuals adjusted for the signal phase at baseline. Circles = individual data points; black horizontal line = across participants mean; grey box = ± standard error of the mean across participants.

$[F_{(1, 70)}=0.661, p=0.419]$ and no three-way interaction pupil time window x adjustment x group $[F_{(2, 140)}=2.43, p=0.092]$.

As these evoked responses were correlated, it is possible that one of them is mediating the effect of the other on reaction time. We tested this using multiple regression models with reaction time as dependent variable and CNV and/or PD residuals adjusted for the signal phase at baseline as independent variables. We were interested in understanding the cause of the behavioral variability observed. Therefore, we focused on the neural responses that occur before the button press and only included in these analyses the pupil dilation response measured during the preparatory period before target onset (measured at the same time window as the CNV). The reaction time variance explained by the model including CNV and pupil residuals was significantly higher than the variance explained by the model including only the pupil dilation residuals or the model including only the CNV residuals [*Figure 12C*; $R^2$ across participants mean ± SD: pupil only model young group = 2% ± 3%; pupil only model older group = 2% ± 3%; CNV only model young group (average across channels)=2% ± 2%; CNV only model older group (average across channels)=4% ± 4%; pupil +CNV model young group (average across channels)=4% ± 3%; pupil +CNV model older group (average across channels)=7% ± 4 %]. In repeated measures ANOVA with model as within-subject factor (three models: pupil only, CNV only, and pupil and CNV together) and group as between-subject factor, the across channels average correlation coefficients showed a significant effect of model $[F_{(2, 140)}=33.5, p<0.001]$, no group x model interaction $[F_{(2, 140)}=3.06, p=0.050]$ and a significant group effect $[F_{(1, 70)}=6.14, p=0.016]$ with the older group presenting higher $R^2$ values. Post hoc tests revealed that the CNV and pupil

model had significantly higher $R^2$ in comparison with the pupil only model and the CNV only model (ps <0.001), while the reaction time variances explained by the pupil only and CNV only models were not significantly different (p=0.056). These findings suggest that the pupil dilation response and the CNV contribute independently towards behavior variability.

## Pre-stimulus pupil size and EEG fluctuations are associated with reaction time

Our results suggest that the effect of the slow ongoing fluctuations on the evoked responses is not behaviorally relevant. However, it is possible that these signal fluctuations affect behavior independently from their effect on the evoked responses. We tested this hypothesis using within-subject regression analyses including as independent variables the cosine and sine of the signals phase at cue onset and reaction time as dependent variable. We fitted separate models for the EEG and pupil signals. In the EEG, the estimated coefficients for the sine function were significantly different from zero in two left temporal channels (FT7 and T7) after controlling for multiple comparisons. The coefficients were negative suggesting that faster reaction times occurred during the descending phase of these slow fluctuations. The estimated coefficients for the cosine function were not significantly different from zero. The coefficients were also not significantly different across groups, after controlling for multiple comparisons. In contrast, in the pupil analyses, the estimated coefficients were significantly different from zero for the cosine function, that captures fluctuations in amplitude and is equivalent to pupil size at baseline [one-sample $t$-test including all participants - $t_{(72)}$ = –3.30, p=0.002]. The sine function estimated coefficients were not significantly different from zero [$t_{(72)}$=–0.863, p=0.391]. Neither set of coefficients showed a significant effect of group [independent samples $t$-tests - cosine: $t_{(71)}$ = –0.449, p=0.655; sine: $t_{(71)}$ = 0.546, p=0.587]. *van den Brink et al., 2016* showed that the effect of pupil baseline on reaction time could be accounted for by the effect of time-on-task (*van den Brink et al., 2016*). We therefore repeated this analysis now controlling for the effect of time-on-task, yet, this relationship between the cosine values and reaction time was still significant [one-sample $t$-tests including all participants - $t_{(72)}$ = –3.70, p<0.001]. In summary, at baseline (before cue onset), the descending phase of the slow signal fluctuations in the EEG in left temporal channels and bigger pupils were associated with faster responses.

## Discussion

Task-related brain evoked responses occur on top of ongoing signal fluctuations that markedly affect their amplitude on a trial-by-trial basis. Our results revealed that as a consequence of older people showing reduced amplitude in the ongoing slow signal fluctuations measured in the EEG and pupil signals, trial-by-trial variability in their evoked responses is also decreased. Once the difference in ongoing signal dynamics was adjusted for, the evoked responses were equally variable in young and older adults. This finding is consistent with the fact that behavioral variability does not decrease with age as would be expected if the evoked responses were more consistent across trials. In fact, although ongoing signal fluctuations strongly affected the amplitude of the evoked responses, these amplitude modulations did not appear to have behavioral consequences.

There are two possible ways through which the slow fluctuations observed in the ongoing signals impact the evoked responses. One possibility is that these slow signal fluctuations reflect fluctuations in brain state that affect the way the brain responds to external stimuli as has been suggested for example for the impact of infra-slow oscillations observed in EEG (*Monto et al., 2008*) or pre-stimulus activity levels in resting-state networks in fMRI (*Boly et al., 2007*). The other possibility assumes that the effect of the ongoing fluctuations on the evoked responses is purely additive and simply reflects the fluctuating baseline and is therefore not relevant for the type of task studied. This second hypothesis is supported by previous fMRI studies in humans (*Fox et al., 2006*; *Schölvinck et al., 2012*) as well as studies of neuronal responses measured in the sensory cortices of mice and cats (*Schölvinck et al., 2015*; *Shimaoka et al., 2019*). Our results further suggest that this latter case dominates the effect of the ongoing EEG and pupil signals on their respective evoked responses, highlighting an effect of ongoing signal's phase: if the ongoing signal is increasing in amplitude, then the evoked response shows a more positive amplitude and vice-versa. Indeed, similar baseline amplitude levels at different signal phases (with amplitude increasing or amplitude decreasing) have different effects

on the amplitude of the evoked responses (see *Figure 9*) and the variance of the evoked responses explained by the models tested increased substantially by including the ongoing signal phase [r • sin(θ)] at cue-onset in comparison with including only the signal amplitude (as captured by the cosine function). Importantly, when we adjusted for the effect of the ongoing fluctuations on the evoked response's amplitude, the correlation between the evoked response and reaction time was at least as strong as without this adjustment, suggesting that the effect of the ongoing signal fluctuations on the evoked responses has no impact on behavior. Interestingly, in both the pupil and EEG signals adjusting for ongoing signal fluctuations unmasked a relationship between the evoked responses amplitude and reaction time that, without the adjustment, was not detectable in the pupil and was evident only in a restricted number of channels in the EEG. This observation suggests that the additional variability imposed on the evoked responses by the ongoing signal can mask brain-behavior associations and therefore should be attended to and adjusted for.

Although, the effect of the ongoing signal on the amplitude of the evoked responses was not behaviorally relevant, ongoing signal fluctuations might reflect changes in brain activity that affect task performance independently from their impact on the evoked responses. In fact, *Schölvinck et al., 2012*, using fMRI data, found that spontaneous activity was associated with visual perception without significantly modulating visual evoked responses. Moreover, an effect of baseline (pre-stimulus) slow activity fluctuations on task performance has been observed in studies using EEG or pupil data (*Gilzenrat et al., 2010*; *Monto et al., 2008*). Accordingly, we explored the relationship between pre-stimulus slow brain signal fluctuations and reaction time, however, it is important to note that as we were using a cued task our pre-stimulus activity refers to activity before cue onset that is at least 1.5 s before target onset. Under these conditions, pre-cue slow signal fluctuations in the EEG were associated with reaction time in two left temporal channels. We found that in these EEG channels the sine of the phase of the ongoing signal was negatively related to reaction time suggesting that faster reactions happen during the descending phase of the ongoing slow EEG fluctuations. In the pupil data, we observed a significant negative relationship between pupil size (captured by the cosine of the signal phase) and reaction time suggesting that larger pupils at cue onset are associated with faster reactions. Large pupils might reflect higher levels of arousal and therefore better task engagement. In fact, this is also what we observe in our pupil evoked response – the larger the pupil response the faster the reaction times. The elucidation of the mechanisms by which baseline pupil size affects task performance is hampered by the fact that baseline pupil is intrinsically linked to the phase of the ongoing slow fluctuations that in turn affect the amplitude of the evoked responses in an additive manner and, in this case, in the opposite direction from the one predicted from its effect on behavior: large baseline pupils are associated with fast reaction times but are also associated with smaller pupil evoked responses that are in turn associated with slower reaction times. This observation further supports the idea that the effect of pupil baseline on behavior is independent of the effect of the pupil baseline on the evoked responses.

The association between baseline signal fluctuations and reaction time was not significantly different across groups suggesting that signal fluctuations affect reaction time similarly in young and older people. Larger signal fluctuations in the young group would suggest higher reaction time variability. This is not the case, as young participants display lower reaction time variability. It is therefore unlikely that this mechanism contributes towards the age-related differences in behavioral variability.

Fluctuations in pupil size under constant illumination conditions reflect activity in a restricted number of brain regions. The main areas that have been identified in non-luminance modulation of pupil size are the superior colliculus and the locus coeruleus (LC; *Joshi and Gold, 2020*). The involvement of the superior colliculus in pupil dilation responses is thought to predominate during the orienting response to sensory stimuli (*Wang et al., 2014*), while activation of the LC modulates pupil size during resting periods as well as during cognitive arousal (*Joshi et al., 2016*; *Murphy et al., 2014*; *Reimer et al., 2016*). However, the pupil inverse problem (which brain areas are being modulated when we observe changes in pupil size) remains to be solved (*Joshi and Gold, 2020*). Our results suggest that the fluctuations observed in the ongoing pupil signal and the pupil evoked responses correlate with reaction time even though the effect of the ongoing signal on the evoked responses was not behaviorally relevant. The component of the ongoing pupil signal associated with task performance might be separate from the large fluctuations observed and might even correspond to the same mechanism that is further activated in response to the cue. It is likely that the component of the

ongoing pupil signal that is behaviorally relevant constitutes a small part of the signal. In fact, studies describing the association between neural activity and ongoing pupil dilation events report significant but small correlations suggesting that a large proportion of the pupil variance remains unexplained (*Joshi et al., 2016*; *Megemont et al., 2022*; *Reimer et al., 2016*).

The idea that the ongoing pupil signal fluctuations and the task-related pupil evoked response can be two largely independent signals, challenges previous claims. The prevailing model suggests that high baseline (tonic) activity in the LC is associated with weaker task-related (phasic) responses, and that pupil data parallels this observation (*Aston-Jones and Cohen, 2005*; *Gilzenrat et al., 2010*). However, the application of this model to human pupil data acquired during performance of typical psychophysical tasks has been challenged given the relatively small modulations in arousal that occur under these conditions (*Joshi and Gold, 2020*). Our analyses further suggest that the effect of pupil baseline on the pupil dilation responses is not behaviorally relevant and merely additive.

Previous fMRI studies suggest that task evoked responses are affected by ongoing signal fluctuations occurring in the same brain areas but that these ongoing fluctuations are shared across brain regions (*Fox et al., 2006*; *Schölvinck et al., 2012*). Similarly, single-cell electrophysiology and optical imaging with voltage sensors in anesthetized cats suggest that sensory responses in individual neurons are variable from trial-to-trial and that this variability reflects the sum of the sensory response with an ongoing signal shared across neurons (*Arieli et al., 1996*; *Schölvinck et al., 2015*). The CNV preparatory potential originates not from an unitary neural mechanism but from a set of brain areas, including the supplementary motor area (SMA), anterior cingulate cortex (ACC) and the thalamus (*Nagai et al., 2004*). Likely, these areas will display ongoing activity that will sum to the evoked responses. Nevertheless, the ongoing EEG fluctuations studied probably originate in a large set of brain areas that will include, but not be limited to, the brain areas involved in the preparatory mechanisms linked to the CNV potential.

The two markers of preparatory neural activity, the CNV and the pupil dilation response, were found to be correlated on a trial-by-trial basis. The correlation coefficients were stronger after adjusting for the slow ongoing fluctuations of each signal, further highlighting the importance of adjusting for these signal fluctuations. A correlation between the CNV and the pupil dilation response had already been suggested before on between-subjects correlation analysis (*Hepsomali et al., 2019*). We found that the correlation was stronger when comparing the CNV amplitude with the pupil dilation measured with a 500 ms delay in line with the idea that the pupil response is slow and lags the associated cortical events by hundreds of milliseconds (*Reimer et al., 2016*). A likely brain area linking the CNV with the pupil dilation response is the anterior cingulate cortex (ACC). Activity in the ACC is observed during preparatory periods associated with the CNV potential (*Nagai et al., 2004*) and it is associated with pupil dilation events (*Joshi et al., 2016*; *Joshi and Gold, 2022*). Nevertheless, it is intriguing to note that the correlation between the CNV and the pupil dilation response presented a clear posterior topography challenging the focus solely on the ACC. The fact that the CNV and pupil dilation independently explain some of the reaction time variance suggests the existence of brain systems that contribute towards the CNV amplitude independently of the brain systems associated with pupil dilation. Given that, without adjusting the CNV for the effect of the ongoing signal fluctuations, the correlation between the CNV and reaction time was observed mostly in left frontocentral channels, it is likely that this reflects the effect of a component of the CNV associated with motor preparatory activity independent of the orienting response. In turn, it is also intriguing that pupil dilation contributes towards reaction time independently of its effect on the CNV. Possible mechanisms are the priming of sensory cortices or the speeding up of the communication between sensory and motor cortices by the ascending brainstem neuromodulatory systems (*Vazey et al., 2018*).

Our analyses showed that ageing affects the dynamics of the ongoing EEG signal (reflecting cortical activity) and the ongoing pupillary signal (reflecting activity in brainstem neuromodulatory systems). In both signals, older people presented flatter spectra with reduced power in the slower aperiodic fluctuations. This flattening of brain signal spectra with ageing has been described before in EEG and ECoG data (*Voytek et al., 2015*; *Waschke et al., 2017*), is also evident in MEG data (*Vlahou et al., 2014*), and is here reported in the pupil signal suggesting an age-related decrease in the amplitude of the slow fluctuations in cortical and subcortical networks. In line with our findings, *Vlahou et al., 2014* showed that, in MEG data, the power of slow fluctuations decreases linearly with ageing (*Vlahou et al., 2014*). Sequential activation and de-activation of resting-state networks is observed in the

ongoing EEG data both during rest and task performance (*Abreu et al., 2020*; *Liu et al., 2017*). This ongoing dynamic possibly underlies the ongoing signal fluctuations we observe at the channel level EEG. The fact that EEG, MEG and pupil spectra decrease in amplitude with increasing frequency suggests that brain activation patterns that involve large activation volumes accumulate relatively slowly (the larger the brain volume recruited the slower the activity changes). Ageing is characterized by a significant decrease in the strength of long-range structural connections (*Westlye et al., 2010*) as well as a decrease in the functional connectivity (correlated BOLD activation patterns) within large-scale brain systems, like the default mode network (*Sala-Llonch et al., 2015*), consistent with the idea that simultaneous activation of distant brain areas supported by long-range structural connectivity is impaired in older individuals (*Andrews-Hanna et al., 2007*). Reduced amplitude of slow EEG fluctuations might therefore be related to impaired recruitment of large-scale brain systems.

We started this study with the assumption that if behavior is more variable in older people, then the neural responses that underlie that behavior should also show increased variability. We found that after controlling for differences in the ongoing brain signals, task-related evoked responses were equally variable in young and older adults. This leaves unanswered the origin of the increased behavioral variability observed in older people. It is likely that behavioral variability emerges from the summation of variability at different stages of task processing and thus it is not all captured by the preparatory responses studied here. Moreover, in our study, behavioral variability was only slightly increased in the older group and that might have hindered the detection of increased variability in the evoked responses. The age-related increase in reaction time variability depends on the task used to measure it (*Dykiert et al., 2012*). For example, the age difference in reaction time variability is smaller in simple RT tasks than in choice tasks and is smaller in tasks using longer preparatory intervals as the ones used in our task (*Dykiert et al., 2012*). Future studies addressing this question would benefit from using a behavioral task where age differences in behavioral variability are more pronounced.

## Conclusions

The current study shows that the amplitude of the preparatory evoked potential, the CNV, and the preparatory pupil dilation responses show comparable trial-by-trial variability in young and older adults once the effect of the ongoing signal fluctuations is considered. This finding suggests a dissociation between ongoing brain activity, which presents reduced variability in older adults, and task-related responses, which do not. This dissociation might explain why although ongoing brain activity is less variable in older adults, behavior variability increases with ageing.

## Materials and methods

### Participants

Thirty-six young adults (mean age ± SD = 23±3 years; 29 women; 3 left-handed) and thirty-nine older adults (mean age ± SD = 60±5 years; 31 women; 3 left-handed) were included in this study. Participants' characteristics were reported elsewhere (*Ribeiro and Castelo-Branco, 2019a*). EEG data from one older participant and pupil data from one young adult and one older adult with light-coloured eyes were not included due to poor data quality.

The study was conducted in accordance with the tenets of the Declaration of Helsinki and was approved by the Ethics Committee of the Faculty of Medicine of the University of Coimbra (reference number CE-002–2016). Written informed consent was obtained from the participants, after explanation of the nature and possible consequences of the study.

### Task design

The task was designed and run with the Psychophysics Toolbox, version 3 (*Brainard, 1997*), for Matlab (The MathWorks Company Ltd). Task design is schematized in *Figure 1* and has been described in detail elsewhere (*Ribeiro and Castelo-Branco, 2019a*). Briefly, we applied two cued auditory tasks: a cued simple RT task and a cued go/no-go task. In this study, for the sake of conciseness, we report only the analyses of the go/no-go task data. Equivalent results were obtained with the data from the simple RT task with the exception that PD amplitude, without the adjustment for the slow ongoing fluctuations, correlated with reaction time. Nevertheless, after the adjustment the correlation was significantly stronger. The auditory stimuli used were three different pure tones, suprathreshold, with

duration of 250ms, with the following frequencies: cue - 1500 Hz; go stimulus - 1700 Hz; no-go stimulus - 1300 Hz. Participants performed the two tasks, cued simple RT and cued go/no-go, sequentially. The order of the tasks was counterbalanced across participants, that is, half of the participants started with the simple RT and the other half with the go/no-go. In the simple RT task, the cue was followed by the go stimulus (100 trials) to which participants were instructed to respond by pressing a keyboard key as fast as possible with their right index finger. In the go/no-go task, the cue was followed or by the go stimulus (80 trials) or by the no-go stimulus (20 trials). Participants were instructed to respond as fast as possible to the go stimulus with their right index finger, while refraining from responding to the no-go stimulus. The intertrial interval was variable with a median of 7.6 s (min 6.7 and max 19.6 s). The interval between the cue and the target stimuli and between target and the beginning of the next trial were drawn from a nonaging distribution, $-W*\ln(P)$, where W is the mean value of the interval distribution and P is a random number between 0 and 1 (*Jennings et al., 1998*). In our task design, the cue-target interval was $1.5-0.25*\ln(P)$ and the interval between the target and the beginning of the next trial (cue) was $5.2-1*\ln(P)$ in seconds.

In the analysis of task performance, we assessed reaction time and task accuracy. We considered as error trials all trials where the participants responded after cue presentation, failed to respond to the go stimulus (misses), responded to the go stimulus too slowly (slower than 700ms), or responded to the no-go stimulus in the go/no-go condition. These trials were signalled with a feedback tone warning the participants that an error was committed. Statistical analyses of number of errors were performed using nonparametric Mann-Whitney tests.

## EEG data acquisition and pre-processing

As previously described (*Ribeiro and Castelo-Branco, 2019a*), the EEG signal was recorded using a 64-channel Neuroscan system (Compumedics EUROPE GmbH) with scalp electrodes placed according to the International 10–20 electrode placement standard, with reference between the electrodes CPz and Cz and ground between FPz and Fz. Acquisition rate was 500 Hz. Vertical and horizontal electrooculograms were recorded to monitor eye movements and blinks. The participants' head was stabilized with a chin and forehead rest to record pupillographic data simultaneously. Consequently, the electrodes on the forehead, FP1, FPz, and FP2, displayed signal fluctuation artefacts due to the pressure on the forehead rest. These were excluded from the analyses. A trigger pulse was generated at the onset of each stimulus and at every button press. EEG data analysis was performed with the EEGLAB toolbox versions 14.1.1 and 19.1 (*Delorme and Makeig, 2004*) and Matlab custom scripts.

We used independent component analysis (ICA) to eliminate nonbrain artefacts from the data, as described previously. The data, re-referenced to linked earlobes, was then band pass filtered with cut off frequencies of 0.1 and 35 Hz, and periods containing further artefacts were manually removed.

For visualization of the scalp topographies of the EEG analyses results, we used perceptually uniform colour maps developed by *Crameri et al., 2020*.

## Pupil data acquisition and pre-processing

As described previously (*Ribeiro and Castelo-Branco, 2019a*), the pupil diameter of the right eye was measured by an infrared eye-tracker (iView X Hi-Speed 1,250 system from SMI) with a sampling rate of 240 Hz. Analysis of pupil data was performed using Matlab custom scripts and the EEGLAB toolbox in Matlab. Artefacts and blinks were corrected using the blink correction Matlab script by Greg Siegle (stublinks.m) available in http://www.pitt.edu/~gsiegle/ (*Siegle et al., 2003*). Briefly, artefacts, including blinks, were identified as large changes in pupil dilation occurring too rapidly to signify actual dilation or contraction. Linear interpolations replaced artefacts throughout the data sets. Data were smoothed using a 3-point unweighted average filter applied twice.

For each run, pupil data was expressed as the percentage of the mean across the whole run by dividing the run data by the mean pupil size within that run. Pupil data were then cut into epochs locked with the onset of the auditory cue from 200ms before cue onset up to 6 s after. Epochs were baseline corrected by subtracting the average amplitude within the 200ms before cue onset.

Pupil epochs were visually inspected for artefacts not adequately corrected by the automatic linear interpolation procedure. Epochs with remaining artefacts were manually rejected. In addition, we discarded all error trials and correct trials that immediately followed error trials.

## Single-trial amplitude and variability of evoked responses in the EEG and pupil signals

The preparatory evoked responses in the EEG and pupil signals were studied in cue-locked epochs with average baseline amplitude (200ms time window before cue onset) subtracted. The numbers of included trials did not differ significantly across groups (mean ± SD of number of trials included: EEG - young group = 72 ± 7; older group = 71 ± 7; pupil - young group = 66 ± 10; older group = 62 ± 12). Single trial amplitude was calculated by averaging the cue-locked responses within the time window from 1 s to 1.5 s after cue-onset. This time window was chosen to include the period of highest amplitude of the preparatory response while avoiding any activity related to target processing and therefore positioned just before the earliest target onset (1.5 s after cue onset). We chose a 500ms window as a compromise between averaging over time and therefore reducing the 'noise' associated with single trial responses and being close enough to the onset of the go stimulus.

Group comparison of average CNV amplitude and across trials SD of CNV amplitude was performed, for each channel, with independent samples permutation test based on a $t$-statistic. The 'tmax' method was used for adjusting the p values of each variable for multiple comparisons, across the 59 electrodes analysed (*Blair and Karniski, 1993*; *Groppe et al., 2011*).

## Correlation between single trial amplitude of evoked responses and reaction time

For within-subject correlation analyses between reaction time and single-trial amplitude of the evoked responses, we used the Pearson's skipped robust correlation (*Wilcox, 2012*), as implemented in the robust correlation Matlab toolbox (*Pernet et al., 2012*; *Rousseeuw, 1984*). Skipped correlations minimize the effects of bivariate outliers by considering the overall structure of the data. Notably, Pearson's skipped correlation is a direct reflection of Pearson's $r$. At the group level, correlations were considered significant if the correlation coefficients were significantly different from zero. For the EEG, this was established using one sample permutation test based on a $t$-statistic with the 'tmax' method used for adjusting the $p$-values of each variable for multiple comparisons (across the 59 electrodes analysed) (*Groppe, 2021*; *Groppe et al., 2011*).

## Spectral analyses of ongoing signals

EEG and pupil power spectral densities (PSDs) estimated via the Welch's method were fit using the FOOOF algorithm (version 1.0.0) using the Matlab wrapper (*Donoghue et al., 2020*).

In the EEG analyses, PSDs were estimated in single trials in 3.5 s long pre-cue epochs. PSDs were then averaged across trials for each electrode and fit using the FOOOF algorithm with the following settings: peak width limits = [1 8]; peak threshold = 1; aperiodic mode = 'fixed'; minimum peak height = 0.1; max number of peaks = 4; frequency range = [1 35]. Peaks with peak frequency between 7 and 14 Hz were considered within the alpha range and peaks with peak frequency between 14 and 30 Hz were considered within the beta range. Peaks outside these frequencies were rarely detected and were not analysed.

In the pupil analyses, PSDs were estimated in 20 s long epochs from a 4 min recording acquired at the beginning of the experiment during which the participants were passively fixating and listening to the cue stimulus being presented with the same frequency as in the cued auditory tasks but without any overt task (*Ribeiro and Castelo-Branco, 2019a*). The 20 s epochs were cut sequentially and were not locked to the auditory stimulus. These long epochs allowed us to assess the slow signal fluctuations typical of the pupillary recordings. Pupil PSDs were then averaged across trials and were fit using the FOOOF algorithm with the following settings: peak width limits = [-inf.25]; peak threshold = 3.5; aperiodic mode = 'fixed'; minimum peak height = 0.2; maximum number of peaks = 4; frequency range = [0.05 6].

FOOOF model fitting quality was estimated from the model $R^2$. For the EEG data, average $R^2$ across electrodes and across participants was 0.99 for the young group and 0.97 for the older group. These $R^2$ values reflected good model fittings, however, it is important to note that they were significantly lower in the older group, in most EEG electrodes (*Figure 5—figure supplement 1*), suggesting that the parameter estimation was not as good in this group. Surprisingly, the model $R^2$ correlated with the estimated values for exponent and offset – lower $R^2$ values were associated with lower exponents and offsets (*Figure 5—figure supplement 1C*). This relationship was evident in both groups of

participants. To ensure that the study conclusions were not driven by differences in fitting quality, we compared the correlation between the variability in the evoked responses and the fitted spectra with the correlation between the variability in the evoked responses and the raw PSD (*Figure 7—figure supplement 1*). We obtained similar results: the power of low frequencies in the aperiodic fitted spectra and in the raw spectra correlated with the evoked responses amplitude variability. The $R^2$ for the fitting of pupil PSDs were not significantly different across groups ($R^2$ young = 0.99; $R^2$ older = 0.98; *Figure 6—figure supplement 1*). Pupil $R^2$ correlated with the estimated PSD offset values but not with the PSD exponent.

## Estimation of the phase of the slow fluctuations in the EEG and pupil signals

The instantaneous phase and amplitude envelope of the slow fluctuations was estimated using the Hilbert transform. Before applying the Hilbert transform, the pre-processed EEG signal was bandpass filtered between.1 Hz and 2 Hz and the pre-processed pupil signal was bandpass filtered between.1 and.9 Hz. Phase (θ) and amplitude envelope (r) at the time of cue-onset were extracted and used to calculate two variables that represent the circular phase of the signal in Cartesian coordinates: $r \bullet sin\ \theta$ and $r \bullet cos\ \theta$ (*Nurhab et al., 2017*). These variables were included as predictors in the linear regressions studying the effect of pre-stimulus phase angle on the amplitude of the evoked responses. Note that $r \bullet cos\ \theta$ is equal to the amplitude of the filtered signal at each time point.

## The effect of pre-stimulus variables on single trial amplitude of evoked responses

We run within-subject generalized linear regression models, with the single-trial amplitude of the cue-locked evoked responses (average within the time interval from 1 to 1.5 s after cue-onset) as response variable and including run as categorical predictor (two sequential runs were acquired per participant), and, as continuous predictors, time-on-task (defined as the trial number within each run), alpha power (estimated from the FOOOF model fitting of 3.5 s pre-stimulus epochs in electrode POz where it showed the highest amplitude; when no alpha peak was detected, alpha power was set at zero), r • sin θ and r • cos θ (estimated as described above for each signal).

The models explained a large amount of variance of the evoked responses amplitude, however, the explained variance was higher in the young than in the older group [independent samples *t*-test comparing the models'$R^2$ of both groups: EEG $t_{(72)}$ = 2.11 p=0.039; pupil $t_{(70)}$ = 2.88, p=0.005].

## Adjusting single-trial amplitude of the evoked responses for pre-stimulus phase angle of ongoing signals

To adjust the evoked responses amplitude for the effect of ongoing signal fluctuations, we regressed out the effects of the phase of the signal at cue-onset by taking, for each participant, the residuals from the multiple linear regression with response variable the evoked responses amplitude and r • sin θ and r • cos θ as predictors. The regression was run separately for each signal (EEG and pupil).

## Measurement of reaction time variance explained by pupil dilation and/or the CNV responses

To determine if the amount of reaction time variance explained by the CNV was independent from the variance explained by the pupillary response, we run, for each participant, three multiple regression models using the regress.m function in Matlab. We included reaction time as the dependent variable and as independent variables the CNV amplitude residuals and/or the pupil dilation amplitude residuals both adjusted for the pre-stimulus phase angle of the respective ongoing signals, as described above. In model 1, we used as independent variable the pupil dilation residuals, in model 2, we used the CNV amplitude residuals and in model 3, we used both as independent variables. For the models including the CNV amplitude, we run separate models for each EEG channel and took the average $R^2$ across all EEG channels as a measure of the amount of variance explained. We compared the $R^2$ of the three models using a repeated measures ANOVA.

## Software for statistical analyses

Nonparametric Mann-Whitney tests and repeated measures ANOVAs were run using IBM SPSS Statistics software (IBM Corp. Released 2020. IBM SPSS Statistics for Windows, Version 27.0. Armonk, NY: IBM Corp). All the other statistical tests were run using Matlab.

## Additional information

### Funding

| Funder | Grant reference number | Author |
| --- | --- | --- |
| Fundação para a Ciência e a Tecnologia | FCT/UIDB&P/4950/2020 | Miguel Castelo-Branco |
| Fundação para a Ciência e a Tecnologia | PTDC/PSIGER/30852/2017 | Miguel Castelo-Branco |
| Fundação para a Ciência e a Tecnologia | DSAIPA/DS/0041/2020 | Miguel Castelo-Branco |
| Fundação para a Ciência e a Tecnologia | EXPL/PSI-GER/0349/2021 | Maria Ribeiro |

The funders had no role in study design, data collection and interpretation, or the decision to submit the work for publication.

### Author contributions

Maria Ribeiro, Conceptualization, Data curation, Formal analysis, Investigation, Methodology, Project administration, Resources, Software, Visualization, Writing – original draft, Writing – review and editing; Miguel Castelo-Branco, Funding acquisition, Resources, Supervision, Writing – review and editing

### Author ORCIDs

Maria Ribeiro http://orcid.org/0000-0001-6422-3279
Miguel Castelo-Branco http://orcid.org/0000-0003-4364-6373

### Ethics

Human subjects: The study was conducted in accordance with the tenets of the Declaration of Helsinki and was approved by the Ethics Committee of the Faculty of Medicine of the University of Coimbra (reference number CE-002-2016). Written informed consent, and consent to publish, was obtained from the participants, after explanation of the nature and possible consequences of the study.

### Decision letter and Author response

Decision letter https://doi.org/10.7554/eLife.75722.sa1
Author response https://doi.org/10.7554/eLife.75722.sa2

## Additional files

### Supplementary files
• Transparent reporting form

### Data availability

The dataset used in this study has been made available in Open Neuro Repository under the https://doi.org/10.18112/openneuro.ds003690.v1.0.0. All analyses and figures presented in the manuscript resulted from analyses of this dataset. Matlab code used in the analyses is available on GitHub https://github.com/CIBIT-ICNAS/2021_Ribeiro_BrainVariability_Aging, (copy archived at swh:1:rev:e26e2e2ca77e95bdd1db48cb850273e6fd2c9940).

The following previously published dataset was used:

| Author(s) | Year | Dataset title | Dataset URL | Database and Identifier |
|-----------|------|---------------|-------------|-------------------------|
| Maria JR | 2021 | EEG, ECG and pupil data from young and older adults: rest and auditory cued reaction time tasks | https://openneuro.org/datasets/ds003690 | OpenNeuro Repository, 10.18112/openneuro.ds003690.v1.0.0 |

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
