## [Editor Report]

This paper examines an overlooked issue in research on the neurophysiology of cognitive aging – that the brain signals of older and younger subjects may differ due to physiological changes that bear little or no relevance to the cognitive processes being studied. Here, the authors demonstrate that older subjects exhibit reduced slow oscillations in EEG and pupil power and that controlling for these differences eliminated group differences in ERP and pupil dilation variability while strengthening their correlation with reaction time. The work should be of interest to those using EEG to study inter-individual and inter-group differences, particularly in the realm of aging and age-related disorders.

---

## [Decision Letter]

**Decision letter after peer review:**

Thank you for submitting your article "Slow fluctuations in ongoing brain activity decrease in amplitude with ageing yet their impact on task-related evoked responses is dissociable from behaviour" for consideration by *eLife*. Your article has been reviewed by 3 peer reviewers, including Redmond G O'Connell as the Reviewing Editor and Reviewer #1, and the evaluation has been overseen by Joshua Gold as the Senior Editor. The following individual involved in review of your submission has agreed to reveal their identity: Marieke Schölvinck (Reviewer #3).

Essential revisions:

1) The authors should provide more detailed behavioural results including false alarms and miss rates.

2) The authors should use more cautious language when discussing the strength and statistical significance of their results in a manner proportionate to the statistical power of this study.

3) In addition to controlling for their influence in the statistical models, the authors should report the degree to which the observed slow EEG/pupil fluctuations correlate with RT variability.

4) The authors should consider the possibility that the left-focussed ERP-RT relationships in fact reflect motor preparation as opposed to the CNV.

*Reviewer #1 (Recommendations for the authors):*

The authors report in the abstract and elsewhere that their study resolves the apparently discrepant observation that older adults have higher RT variability but lower brain signal variability. The reported analyses suggest that, after controlling for background slow oscillations, older and younger groups have equivalent levels of brain signal variability. If so, this still leaves a discrepancy – older adults have higher RT variability despite equivalent brain signal variability. The authors should address this in their Discussion. Might the answer lie in the observation that older and younger adults did not differ in their coefficients of variation?

Insufficient detail is provided on the analysis of the pupil data in the Methods section. How were pupil dilations measured? This is crucial information for the reader to be able to judge the degree to which the analyses might partly reflect variations in baseline amplitude.

The initial analyses of the relationship between CNV and RT highlight a markedly left central focus in the topography. It does not seem appropriate to me to refer to such effects as relating to the CNV which is defined by its frontocentral focus. Not all activity in the pre-target window necessarily relates to the CNV process. A notable alternative candidate process here would be motor preparation with larger amplitudes reflecting greater preparation and resulting in faster RTs. It is interesting that a more frontal focus appears in the relationship when controlling for the slow fluctuations, but I think it's important for the authors to be more careful in their use of the term CNV and to discuss alternative explanations of these effects.

I am wondering why the authors did not test for a correlation between the slow EEG/pupil rhythms and RT in their data?

*Reviewer #2 (Recommendations for the authors):*

– I felt that this was an excellent paper, but its conclusions seemed very technical. Apart from knowing that we should take aperiodic signals into account in future research, I am not sure what we have now learned about aging or behavioral variability. But perhaps that's fine.

– The behavioral part of the paper feels very underdeveloped. In the methods it is explained how accuracy was calculated, yet accuracy results are not reported. I think this is quite important, because group differences in reaction time variability can originate from different sources: i) if one group is more cautious this will result in increased variability, slower RTs and higher accuracy, ii) if one group is worse at the task this will result in increased variability, slower RTs and lower accuracy, iii) not mentioning interactions between both (becoming more cautious because performance is worse) or other explanations. The authors try to address this by analyzing SD/M, but the conclusion that "increase in reaction time variability was linked to the slowdown of the responses in the older group" is very weird because RTs are not significantly different. I would think that DDM analyses would be very informative here, but that might be taking it too far (given that this does not seem to be the main focus of the paper).

– Although the sample size is quite considerable for EEG research (and I can imagine data collection for this kind of research to be far from trivial), from a statistical power perspective having only ~36 participants per group amounts to having power to detect effects of roughly.6 (effect size), which is not very sensitive. This clearly plays in the current manuscript as there are several p=.07's (RT differences; PD differences). It is great to see that the authors do no try to interpret these (e.g. not calling the "marginally significant") but at the same time, in my reading much of these.07s are very likely an issue of statistical power (i.e., BFs would be "uninformative"). The same goes for the correlations in table 1: N=38 gives you power to detect r=.43 correlations only, so I am not sure what to make of these n.s. correlations. Also, I guess that contrary to the EEG analyses, no correction for multiple comparisons was applied here?

– P.12, lines 21 and onwards; for this result to be compelling the authors should not just show that these effects become n.s., but that there is a "significant decrease" in effect with and without the adjustment (cf. Nieuwenhuis et al., 2011, Nat Neuro).

*Reviewer #3 (Recommendations for the authors):*

– There are a lot of figures with plots showing differences e.g. in mean reaction time, amplitude of the CNV, etc. It would be helpful to indicate significance on these plots (e.g. by an asterisk), so that one does not have to search in the text whether something is significant or not.

– In the Results, the authors describe that ongoing pupil fluctuations happen at slower time scales than EEG fluctuations, and therefore longer time periods are needed to study their dynamics. While I agree completely with this statement, I am wondering why the pupil recordings from the beginning of data acquisition are analysed, and not from during the task? Wouldn't this be possible too, since there are no visual stimuli in this task?

– The correlation between CNV amplitude fluctuations and behaviour fluctuations (Figure 3) is being described as 'robust' and 'predicted behaviour fluctuations'; however, the correlations are close to zero for both younger and older adults. I would argue for less strong wording here.

---

## [Author Response]

Essential revisions:1) The authors should provide more detailed behavioural results including false alarms and miss rates.

This information is now included in the manuscript (Results section page 6). See also response to reviewer #2.

2) The authors should use more cautious language when discussing the strength and statistical significance of their results in a manner proportionate to the statistical power of this study.

We apologize for not being entirely clear. The word “robust” was used to describe the type of statistical test used and not the results obtained. We have now refrained from using this word in the Results section. Information regarding the robust statistics tests used are detailed in the methods. See response to reviewer #3.

3) In addition to controlling for their influence in the statistical models, the authors should report the degree to which the observed slow EEG/pupil fluctuations correlate with RT variability.

This analysis is now included in the manuscript (Results section pages 28/29; discussion pages 30/31). See also response to reviewer #1.

4) The authors should consider the possibility that the left-focussed ERP-RT relationships in fact reflect motor preparation as opposed to the CNV.

This is now discussed in the manuscript (Results section page 8 and Discussion section page 33). See also response to reviewer #1.

Reviewer #1 (Recommendations for the authors):The authors report in the abstract and elsewhere that their study resolves the apparently discrepant observation that older adults have higher RT variability but lower brain signal variability. The reported analyses suggest that, after controlling for background slow oscillations, older and younger groups have equivalent levels of brain signal variability. If so, this still leaves a discrepancy – older adults have higher RT variability despite equivalent brain signal variability. The authors should address this in their Discussion. Might the answer lie in the observation that older and younger adults did not differ in their coefficients of variation?

We have now included this topic in the discussion of the manuscript (page 34).

Insufficient detail is provided on the analysis of the pupil data in the Methods section. How were pupil dilations measured? This is crucial information for the reader to be able to judge the degree to which the analyses might partly reflect variations in baseline amplitude.

The Methods section has been extended to include further details (page 36/37).

The initial analyses of the relationship between CNV and RT highlight a markedly left central focus in the topography. It does not seem appropriate to me to refer to such effects as relating to the CNV which is defined by its frontocentral focus. Not all activity in the pre-target window necessarily relates to the CNV process. A notable alternative candidate process here would be motor preparation with larger amplitudes reflecting greater preparation and resulting in faster RTs. It is interesting that a more frontal focus appears in the relationship when controlling for the slow fluctuations, but I think it's important for the authors to be more careful in their use of the term CNV and to discuss alternative explanations of these effects.

We agree that these observations suggest that the preparatory response includes a motor preparation component. Our participants were all responding with the right hand independently of being right- or left-handed. Left central cortical anticipatory responses are very likely related to motor preparation processes (contralateral to the response hand), as suggested by the reviewer. This is now included in the Results section, page 8, and Discussion section page 33.

I am wondering why the authors did not test for a correlation between the slow EEG/pupil rhythms and RT in their data?

We did not include these analyses on the original manuscript for the sake of conciseness. Nevertheless, as the reviewers showed interest in them, we have now decided to include them. We found that the ongoing slow signal fluctuations at baseline (pre-cue) were associated with reaction time in both the EEG and pupil signals. This is now presented in the Results section pages 28/29 and discussed in the Discussion section pages 30/31.

Reviewer #2 (Recommendations for the authors):– I felt that this was an excellent paper, but its conclusions seemed very technical. Apart from knowing that we should take aperiodic signals into account in future research, I am not sure what we have now learned about aging or behavioral variability. But perhaps that's fine.

We agree that one of the main contributions of our manuscript relates to the demonstration that ongoing brain activity can have a strong effect on evoked responses and that this modulation should be considered in future research. Importantly, when applied to the question ’does variability in task-related evoked responses change with ageing?’, taking into consideration this confound was crucial to determine that evoked responses are not less variable in older people. This is an important piece of information for the understanding of the origin of increased behaviour variability in ageing, yet, as pointed out by the reviewers, it does not answer what changes with ageing that leads to an increase in behavioural variability, and this is an important question to be addressed in future studies. This is now discussed in the Discussion section page 34.

– The behavioral part of the paper feels very underdeveloped. In the methods it is explained how accuracy was calculated, yet accuracy results are not reported. I think this is quite important, because group differences in reaction time variability can originate from different sources: i) if one group is more cautious this will result in increased variability, slower RTs and higher accuracy, ii) if one group is worse at the task this will result in increased variability, slower RTs and lower accuracy, iii) not mentioning interactions between both (becoming more cautious because performance is worse) or other explanations. The authors try to address this by analyzing SD/M, but the conclusion that "increase in reaction time variability was linked to the slowdown of the responses in the older group" is very weird because RTs are not significantly different. I would think that DDM analyses would be very informative here, but that might be taking it too far (given that this does not seem to be the main focus of the paper).

The parameters of the go/no-go task were such that the task was relatively easy and the number of errors of commission were low and not significantly different across groups. Older people showed a significantly higher number of misses (trials where they did not respond to the go stimuli) and significantly higher number of times where they impulsively responded to the cue. This happened however in a small number of trials. These results have already been presented in our previous paper (Ribeiro and Castelo-Branco 2019) and are now briefly presented in this manuscript (Results section page 6).

We have removed the sentence “increase in reaction time variability was linked to the slowdown of the responses in the older group” as our analyses do not allow for a clear identification of the origin of the increased reaction time variability.

DDM analyses would indeed be informative, however, these are usually applied to choice tasks with reaction time data for both choices (not the case for our go/no-go task). Also, we have a relatively small number of trials to fit these models.

– Although the sample size is quite considerable for EEG research (and I can imagine data collection for this kind of research to be far from trivial), from a statistical power perspective having only ~36 participants per group amounts to having power to detect effects of roughly.6 (effect size), which is not very sensitive. This clearly plays in the current manuscript as there are several p=.07's (RT differences; PD differences). It is great to see that the authors do no try to interpret these (e.g. not calling the "marginally significant") but at the same time, in my reading much of these.07s are very likely an issue of statistical power (i.e., BFs would be "uninformative"). The same goes for the correlations in table 1: N=38 gives you power to detect r=.43 correlations only, so I am not sure what to make of these n.s. correlations. Also, I guess that contrary to the EEG analyses, no correction for multiple comparisons was applied here?

It is true that such *p* values reflect small effect sizes in those particular comparisons. Also, it is correct that the correlation analyses shown in Table 1 were not corrected for multiple comparisons. We have now, nevertheless, run the correlation analyses including all channels and show that even after correcting for multiple comparisons across channels the relationships are still significant. Note that to include in the manuscript all the figures that were in the supplementary material, we have substituted Table 1 for a figure and its supplement (Figure 7 and Figure 7 – supplementary figure 1. Results section page 15-18).

– P.12, lines 21 and onwards; for this result to be compelling the authors should not just show that these effects become n.s., but that there is a "significant decrease" in effect with and without the adjustment (cf. Nieuwenhuis et al., 2011, Nat Neuro).

We agree that a statistical comparison across conditions is required to establish an effect of adjustment for the confounds. We have now included in the results the interaction effects from the repeated-measures ANOVA comparing CNV variability including as within-subject factor adjustment (with or without adjusting for the aperiodic measures) and group as between-subject factor. Significant adjustment x group interaction effects were observed for all comparisons (Results section, page 18).

Reviewer #3 (Recommendations for the authors):– There are a lot of figures with plots showing differences e.g. in mean reaction time, amplitude of the CNV, etc. It would be helpful to indicate significance on these plots (e.g. by an asterisk), so that one does not have to search in the text whether something is significant or not.

Thank you for this suggestion. We have changed the figures to include the statistical significance.

– In the Results, the authors describe that ongoing pupil fluctuations happen at slower time scales than EEG fluctuations, and therefore longer time periods are needed to study their dynamics. While I agree completely with this statement, I am wondering why the pupil recordings from the beginning of data acquisition are analysed, and not from during the task? Wouldn't this be possible too, since there are no visual stimuli in this task?

Our aim was to ensure that we were describing the dynamics of the ongoing pupil signal not contaminated by task related activity. Therefore, we chose to describe the dynamics of the ongoing pupil signal in a separate recording where the participants were fixating the computer monitor but were not involved in any other task. See reply to reviewer #2 for further details. This is now made clearer in the manuscript.

– The correlation between CNV amplitude fluctuations and behaviour fluctuations (Figure 3) is being described as 'robust' and 'predicted behaviour fluctuations'; however, the correlations are close to zero for both younger and older adults. I would argue for less strong wording here.

The word “robust” was used to describe the “robust” statistics used, skipped-correlations (Pernet et al., 2013). Skipped-correlations remove bivariate outliers by taking into account the overall structure of the data, and Pearson’s skipped correlation is a direct reflection of Pearson’s *r*. This is particularly important when using single trial data prone to outliers. We now realize that this was not entirely clear in the Results section and removed the word “robust” in the description of the results. The robust statistics used are explained in the Methods Section.

Moreover, we now refrain from using the word “predict” when describing the associations found between variables.

Pernet, C. R., Wilcox, R., and Rousselet, G. A. (2013). Robust correlation analyses: false positive and power validation using a new open source Matlab toolbox. 3(January), 1–18. https://doi.org/10.3389/fpsyg.2012.00606